# Probably Approximately Correct Causal Analysis

## Abstract

The discovery and analysis of causal relationships is a foundational problem in artificial intelligence, statistics, epidemiology, economics, and beyond. While elegant theories exist for accurate causal discovery given infinite data, real-world applications are inherently *resource-constrained*. Effective methods for inferring causal relationships from observational data must perform well under finite data and time constraints, where "performing well" implies achieving high, though not perfect accuracy. In his seminal paper *A Theory of the Learnable* (Valiant, 1984), Valiant highlighted the importance of resource constraints in supervised machine learning, introducing the concept of Probably Approximately Correct (PAC) learning as an alternative to exact learning. Inspired by Valiant's work, we propose the Probably Approximately Correct Causal (PACC) Analysis framework, which extends PAC learning principles to the causal field. This framework emphasizes both computational and sample efficiency for established causal methods such as propensity score techniques and instrumental variable approaches. Furthermore, we show that it can provide theoretical guarantees for other widely used methods, such as the Self-Controlled Case Series (SCCS) method, which had previously lacked such guarantees.

## 1 Introduction

The challenge of teaching machines to discern causal relationships rather than mere associations has become a central topic in modern statistics and computer science. This concern extends beyond machine learning, as causality has emerged as a pivotal focus across diverse fields such as medicine, statistics, and economics (Athey & Wager, 2021; Angrist & Pischke, 2008). Understanding hidden causal relationships enables not only predictive tasks, answering "*what*?" questions, but also counterfactual reasoning, answering "*what if*?" questions. A typical example involves using electronic health records (EHRs) to determine whether a newly released drug is causing previously unrecognized adverse effects in certain patients.

Although extensive research has been conducted on methods to uncover causal relationships from observational data (Pearl, 2009; Rosenbaum & Rubin, 1983; Spirtes et al., 2000; Peters et al., 2017), much of this work emphasizes asymptotic performance under the assumption of unlimited data. For example, classical methods such as the Peter-Clark (PC) algorithm (Spirtes et al., 2000) guarantee recovery of the correct Markov equivalence class of the true directed acyclic graph (DAG) in the large-sample limit ($n \to \infty$). However, several studies have shown that the PC algorithm can produce incorrect outputs in finite-sample regimes due to statistical errors in conditional independence testing (Kalisch & Bühlmann, 2007; Spirtes et al., 2000). Moreover, the original PC algorithm provides no explicit guidance on how large sample must be to achieve a desired error rate in finite-sample settings, limiting its practical applicability.

In supervised machine learning, a natural approach to addressing resource constraints is to aim for high, rather than perfect accuracy. As more data and time become available, accuracy can improve accordingly. This idea is formalized in the well-known Probably Approximately Correct (PAC) learning theory (Valiant, 1984), which introduces two parameters, $\epsilon$ and $\delta$, to quantify approximate learning. Specifically, under limited resources, the goal is for a learning algorithm $\mathcal{L}$ to achieve error no greater than $\epsilon$ with probability at least $1 - \delta$. As a result, the required sample size should depend upon $\frac{1}{\epsilon}$ and $\frac{1}{\delta}$, since achieving smaller error demands more data and computation. PAC learning thus provides a foundational framework for balancing the trade-offs among accuracy, confidence, and resource availability.

An analogue to PAC Learning theory for the causal field should preserve the same intuition. In particular, a causal learning algorithm is expected to identify the true relationship or true treatment effect with probably approximately high accuracy. Compared to existing causal frameworks, the new theory should prioritize limited-sample performance over large-sample efficacy. It should also provide guidelines for calculating the required sample size based on desired level of accuracy.

It is worth noting that the concept of sample size calculation is well-established in the context of power analysis for randomized controlled trials (RCTs), particularly in clinical research. This practice has driven significant advances in experimental causal inference and has provided biostatisticians with practical guidance for designing robust and efficient trials (Friedman et al., 2010; Chow et al., 2008). The parallels between parameters such as effect size and significance level in RCTs, and $\epsilon$ and $\delta$ in PAC Learning, are also clearly evident. A similar protocol for sample size will be beneficial for other causal methods as well and, more interestingly, provide approximate guarantees for inference in observational causal tasks. To address these considerations, we propose a unified framework called Probably Approximately Correct Causal Analysis(PACC Analysis), which adapts PAC Learning to causal inquiries under data and time constraints.

Building on this foundation, we explore potential applications of the PACC Analysis framework. In PAC Learning, the theory can be used either to (1) demonstrate that an existing supervised machine learning algorithm is likely to succeed on a specific task (Larsen et al., 2023; Xu & Raginsky, 2023), or (2) motivate the development of a new supervised learning algorithm (Rothfuss et al., 2020; Kumar & et al., 2024). Analogously, PACC Analysis can be applied to (1) validate the effectiveness of an existing causal algorithm, or (2) inspire the creation of new causal algorithms. In this paper, we focus on the first objective, leaving the exploration of the second for future work. To define the success of a causal algorithm, we propose that an algorithm is considered successful if it can reliably identify the true data-generating causal model from a set of competing alternatives that include incorrect or spurious causal components. Occasional errors are allowed but should occur with low probability.

**Contributions.** In this paper, we make the following contributions.

- We emphasize the need for causal analysis models tailored to resource-constrained contexts and motivate the proposed PACC framework, which can provide approximate sample efficiency guarantees for observational causal tasks.

- In Section 2, we derive and formally define the PACC Analysis framework. This framework not only inspires the development of new algorithms but also enriches the theoretical understanding of existing ones.

- In Section 4, we demonstrate the application of PACC Analysis by providing the first theoretical guarantee for a variant of the Self-Controlled Case Series (SCCS) method, establishing it as a valid causal algorithm.

- In Section 5, we show how PACC Analysis can be used to model sample complexity for other causal inference approaches, including propensity score methods and instrumental variables.

## 2 PACC Analysis Framework

### 2.1 PAC Learning Preliminary

We begin with the fundamental components of PAC learning (Kearns & Vazirani, 1994; Kearns et al., 1992):

- A learning algorithm (or learner) $\mathcal{L}$,

- A domain (or instance space) $\mathcal{X}$ consisting of all possible instances,

- A fixed probability distribution $\mathcal{D}$ over $\mathcal{X}$,

- An unknown target concept $c$ from a known concept class $\mathcal{C} \subseteq 2^{\mathcal{X}}$.

Here $2^{\mathcal{X}}$ denotes the power set of $\mathcal{X}$, i.e., the set of all subsets of $\mathcal{X}$. The goal of the learning algorithm $\mathcal{L}$ is to produce a hypothesis $h \in \mathcal{C}$ that closely approximates the unknown target concept $c$. To this end, the learner is provided with a training sample $\mathcal{S} = \{(x_1, c(x_1)), \cdots, (x_n, c(x_n))\}$ consisting of $n$ labeled examples drawn independently and identically distributed (i.i.d.) from the distribution $\mathcal{D}$. After training on $\mathcal{S}$, the learner outputs a hypothesis $h = \mathcal{L}(\mathcal{S}) \in \mathcal{C}$.

We say that $\mathcal{L}$ PAC-learns the concept class $\mathcal{C}$ if, for all $\epsilon > 0$, and $\delta > 0$, there exists a sample size $n = n(\epsilon, \delta)$ such that the following condition holds:

$$\mathbb{P}_{\mathcal{S} \sim \mathcal{D}^n}[error(h) \leq \epsilon] \geq 1 - \delta$$

where $error(h)$ measures the distance between the output hypothesis $h$ and the target concept $c$. A common definition of this error is: $error(h) = \mathbb{P}_{x \sim \mathcal{D}}[h(x) \neq c(x)]$ (Kearns et al., 1992).

## 2.2 Instance Space and Causal Model

In modifying the PAC learning framework to a PAC Causal framework, we draw inspiration from Pearl's Causal Review (Pearl, 2009), where he observed that "such questions are causal questions because they require some knowledge of the data-generating process." For example, a typical causal question in RCTs is whether taking a drug or receiving an exposure $Z$ changes the probability of a subsequent outcome or event $Y$. The corresponding data-generating process can be represented as a joint probability distribution consisting of (i) a probability distribution over patient covariates; (ii) a uniform Bernoulli distribution governing the assignment of treatment $Z$; and (iii) a conditional probability distribution over $Y$, given the covariates and treatment value $Z$. The same causal question can also be investigated using observational data, such as EHRs or health insurance claims. However, in the observational setting, the joint distribution may differ, particularly in the distribution governing treatment assignment.

More generally, the data-generating process in a causal setting can be described using a variety of models, including Bayesian networks, structural equation models, point process models, and randomized Turing machines. Despite their differences, all of these frameworks induce probability distributions over some domain. To unify these representations, we generalize them as causal models over an abstract instance space $\mathcal{I}$, formally defined as follows:

**Definition 1** (Instance Space). *Let the instance space $\mathcal{I}$ be the set of all possible examples considered in a causal problem. Each element $\xi \in \mathcal{I}$ represents a complete configuration of variables relevant to the problem, such as covariates, treatments, and outcomes.*

**Example 1** (Illustrative Instance Spaces). *Several common forms of instance spaces include:*

2.1 *Assignments to $n$ binary variables, represented as $[0, 1]^n$.*

2.2 *Strings over an alphabet of $n$ symbols. For example, the symbols may represent medical event types, and a string may correspond to a patient's sequential medical record.*

2.3 *Same as 2.2, but with real-valued times attached to events.*

2.4 *Assignments to $n$ real-valued variables, represented as $\mathbb{R}^n$, i.e., points in $n$-dimensional Euclidean space.*

**Definition 2** (Causal Model). *A causal model $\mathcal{M}$ over an instance space $\mathcal{I}$ is a synthetic model that captures the underlying data-generating process and induces a probability distribution over $\mathcal{I}$.*

**Example 2** (Illustrative Causal Models). *Examples of causal models corresponding to the instance spaces introduced earlier include:*

4.1 *A Bayesian Network over $n$ binary variables, where the causal structure is represented by a DAG. Each node corresponds to a variable and is associated with a conditional probability table (CPT).*

4.2 *A probabilistic grammar or a randomized Turing machine generating strings over an alphabet of $n$ symbols.*

    *4.3 A marked point process, such as a Hawkes process, a piecewise-constant conditional intensity model, or a neural point process, where event types and temporal dependencies are modeled causally.*

    *4.4 An n-variate Gaussian distribution over $\mathbb{R}^n$, where the covariance matrix captures the dependencies among continuous variables.*

## 2.3 Causal Concept

In the RCT example, although the joint probability distribution may be arbitrarily complex, the specific causal question focuses on whether a direct causal edge exists from the exposure $Z$ to the outcome $Y$. Many causal questions share this property: they do not require recovering the full distribution over the instance space $\mathcal{I}$, but instead aim to identify a specific structural feature or causal relationship within the model. This contrasts with supervised machine learning and PAC learning, where the goal is to recover the entire label-generating mechanism. To formalize this notion, we introduce the concept of a causal concept, which refers to a specific property or feature of a causal model—such as the presence or absence of a causal edge—that we wish to identify.

In the RCT setting, randomized treatment assignment enables causal inference by allowing us to distinguish between a single pair of competing causal models $\langle \mathcal{M}_1, \mathcal{M}_2 \rangle$. These models are identical in all respects except for the presence or absence of a direct causal edge from $Z$ to $Y$, typically accompanied by a corresponding change in the associated parameter. This binary comparison can be extended to observational studies by considering a family of model pairs, where each pair differs only in the existence of the edge from $Z$ to $Y$, while agreeing on all other aspects. These shared aspects may be arbitrarily complex and represented by any valid joint distribution over the instance space. The goal of a causal learning algorithm is to consistently identify the true causal model from competing pairs, even in the presence of challenges such as unmeasured confounding, model misspecification, and distributional variation.

**Definition 3** (Causal Concept). *Let $\mathcal{I}$ be an instance space. A causal concept $c$ is defined as any property that a causal model over $\mathcal{I}$ may or may not possess. It is formally represented by a family of model pairs*

$$\mathcal{F}_c = \{\langle \mathcal{M}_{j1}, \mathcal{M}_{j2} \rangle\}_{j \geq 1},$$

*where for each pair, the model $\mathcal{M}_{j1}$ possesses the property corresponding to concept $c$, and $\mathcal{M}_{j2}$ does not. All other aspects of the models in each pair are held fixed or assumed comparable.*

**Example 3** (Illustrative Causal Concepts). *Examples of causal concepts corresponding to previously discussed causal models include:*

    *6.1 Directed influence in a Bayesian network: Variable A influences variable B; i.e., pairs of Bayesian networks in which the first model includes a directed edge A to B, while the second excludes it. If desired, one may restrict to pairs where the causal effect of A on B exceeds a minimum threshold $\delta$, meaning the probability of B changes by at least $\delta$ when the value of A changes. If parts of the structure are known, only models conforming to that structural constraint are included in the family.*

    *6.2 Sequential dependency in probabilistic grammars: An occurrence of symbol b always immediately follows an occurrence of symbol a in any sequence, e.g., a might represent an ischemic stroke and b the administration of tPA. We consider pairs of probabilistic prefix grammars (Frazier & Page, 1994), where in the first model of each pair, every occurrence of b follows an a on the right-hand side of all production rules, and no production rule deletes any b that immediately follows an a. In contrast, the second model in each pair includes at least one violation of this constraint.*

    *6.3 Excitation in temporal point processes: The occurrence of one event type (e.g., ischemic stroke) increases the rate or likelihood of another event type (e.g., administration of tPA). Pairs of marked Hawkes processes are constructed such that the first model encodes positive influence (shortened time-to-event), while the second encodes no such effect.*

    *6.4 Marginal independence in multivariate Gaussians: Continuous variables A and B are dependent vs. independent. The first model in each pair is a multivariate Gaussian distribution in which A and B have non-zero covariance; in the second model, their covariance is zero.*

In the PAC learning framework, the effectiveness of a learning algorithm $\mathcal{L}$ is influenced by the complexity of the concept class, typically quantified using the VC dimension (Vapnik & Chervonenkis, 2015). In the causal setting, however, additional complexity arises from two key sources: (i) the inherent complexity of individual causal models, and (ii) the degree of similarity between competing causal models.

Regarding the first, PAC learning commonly assumes a specific representation language for target concepts, such as decision trees, Boolean formulae, or deterministic finite automata (DFAs), with an associated notion of representation size (e.g., the number of symbols or bits).[1] By analogy, a natural measure of the size $|\mathcal{M}|$ of a causal model $\mathcal{M}$ could be the total number of bits required to represent the DAG structure and its associated parameters (e.g., CPTs in a Bayesian network). Alternative complexity measures, such as the number of edges or parameters, provide intuitive and computationally convenient proxies for model size, especially in sparse or low-dimensional settings. However, these metrics may not fully capture the descriptive or inferential complexity of a causal model. Their use should therefore be context-sensitive, as they can introduce additional challenges or oversimplify model comparisons.

As for the second aspect, we constrain the family of causal model pairs so that the difference between models in each pair is at least $\delta$. In the simplest case, $\mathcal{F}_{\delta,c}$ contains just one pair $\langle \mathcal{M}_1, \mathcal{M}_2 \rangle$, for example, a Bayesian network $\mathcal{B}$ generating the data and an alternative network $\mathcal{B}'$. In this case, the concept $c$ is simply $\mathcal{B}$ itself, and $\delta$ is a numerical summary of how different $\mathcal{B}'$ is from $\mathcal{B}$. This difference might be measured using Kullback–Leibler (KL) divergence (Cover & Thomas, 1999) between the distributions, or, if $\mathcal{B}$ and $\mathcal{B}'$ share the same structure, it might be the maximum difference between corresponding parameters such as conditional probabilities. More generally, $\delta$ can be expressed as a function over the parameter spaces $\Theta$ and $\Theta'$ of models $\mathcal{B}$ and $\mathcal{B}'$ respectively, i.e., $\delta = f(\Theta, \Theta')$. Our approach is to set a threshold between $0$ and $\delta$ and to test a sample drawn from data against that threshold.

We extend this further by allowing a family $\mathcal{F}_{\delta,c}$ of model pairs instead of just a single pair, in order to account for uncertainty about the true models. For example, $\mathcal{B}$ could be any Bayesian network that includes an edge from variable $X$ to $Y$, without placing constraints on the rest of the network, thus permitting the presence of additional confounding components. We require that the learning algorithm succeeds for every possible pair in the family, and $\delta$ is well-defined for all such pairs. We thus update the notation of the causal family to include this threshold parameter, writing it as $\mathcal{F}_{\delta,c}$.

## 2.4 PACC Analysis

We are now ready to present the formal definition of PACC Analysis in Definition 4, followed by a corresponding general algorithm in Algorithm 1.

**Definition 4.** *For any $0 < \delta$ and $0 < \epsilon < 1$, let $c$ be a target causal concept, and let $\mathcal{F}_{\delta,c} = \{\langle \mathcal{M}_{j1}, \mathcal{M}_{j2} \rangle\}_{j \geq 1}$ denote the corresponding causal family over the instance space $\mathcal{I}$.*

*For a specific pair $\langle \mathcal{M}_{j1}, \mathcal{M}_{j2} \rangle \in \mathcal{F}_{\delta,c}$, the learner $\mathcal{L}$ is given access to a sample $\mathcal{S}$ generated from either $\mathcal{M}_{j1}$ or $\mathcal{M}_{j2}$, and is tasked with identifying which model generated the data. We say that the learner $\mathcal{L}$ PACC-discovers the causal concept $c$ if, for any such pair, it correctly identifies the data-generating model with probability at least $1 - \epsilon$, using a sample of size $|\mathcal{S}|$ that is polynomial in $\frac{1}{\epsilon}$ and $\frac{1}{\delta}$.*

One may observe a superficial similarity between classical hypothesis testing and the PACC Analysis framework, as both rely on statistical tests to distinguish between a null and an alternative hypothesis/causal model. If the causal family contains only a single pair of models, the problem reduces to a standard hypothesis testing task: determining whether the null hypothesis (represented by the second model) can be rejected in favor of the alternative (the first model). However, in the more general case where the causal family contains multiple pairs of models, the problem becomes fundamentally different, taking the form of a "worst-case" analysis. We discuss this further in Section 3.3.

---

[1]Different representations can affect the difficulty of the learning problem, even when describing the same concept. For example, DFAs may require exponentially more states than equivalent NFAs, decision trees and Boolean formulae can vary in compactness, and bit-level encodings often yield longer descriptions than symbolic ones.

---

**Algorithm 1** PACC Analysis General Framework

---

**Requirements** Parameters: $\epsilon \in (0,1)$, $\delta > 0$; a causal concept $c$; and the corresponding causal family $\mathcal{F}_{\delta,c} = \{\langle \mathcal{M}_{j1}, \mathcal{M}_{j2} \rangle\}_{j \geq 1}$.

1: **Input:** Any pair $\langle \mathcal{M}_{j1}, \mathcal{M}_{j2} \rangle \in \mathcal{F}_{\delta,c}$, and a sample $\mathcal{S}$ generated from either $\mathcal{M}_{j1}$ or $\mathcal{M}_{j2}$, where the sample size $|\mathcal{S}|$ is polynomial in $\frac{1}{\epsilon}$ and $\frac{1}{\delta}$.

2: **Evaluation:** The learner $\mathcal{L}$ evaluates the consistency of each model in the pair with the observed sample $\mathcal{S}$.

3: **Output:** The learner $\mathcal{L}$ outputs the model (either $\mathcal{M}_{j1}$ or $\mathcal{M}_{j2}$) that it predicts to be the data-generating model.

4: **Guarantee:** With probability at least $1 - \epsilon$, the output model is the true data-generating process.

---

## 3 Related Work

### 3.1 PAC Learning

After Valiant established the foundation of PAC learning theory (Valiant, 1984), it has grown into a fruitful and vibrant field. In its early development, Blumer et al. (Blumer et al., 1989) showed that a concept class is PAC-learnable if and only if it has a finite Vapnik–Chervonenkis (VC) dimension. Kearns and Valiant (Kearns & Schapire, 1994) further extended PAC learning theory by demonstrating that, under standard cryptographic assumptions, some concept classes are not efficiently learnable despite being PAC-learnable, thereby establishing fundamental computational limits on what can be feasibly learned.

These theoretical insights also inspired a wave of practical algorithmic developments rooted in PAC principles. For example, Schapire's work on the strength of weak learnability (Schapire, 1990) led to the creation of AdaBoost by Freund and Schapire (Freund & Schapire, 1997), which combines multiple weak learners into a strong ensemble. In parallel, PAC learning theory has been successfully extended to specialized domains. For example, PAC-Bayesian methods (McAllester, 1998) integrate Bayesian priors with frequentist generalization guarantees and have been used to derive non-vacuous generalization bounds for deep neural networks (Dziugaite & Roy, 2017). In reinforcement learning, PAC analyses underpin sample-complexity guarantees for algorithms such as R-MAX (Brafman & Tennenholtz, 2002), and later work unified PAC and regret via Uniform PAC bounds (Dann et al., 2017) in finite-horizon Markov decision processes (MDPs).

Building on these developments, the PACC Analysis framework extends PAC learning into the causal domain. This not only broadens the theoretical scope of PAC learning but also introduces new tools and perspectives to causal tasks, especially in resource-constrained, real-world settings.

### 3.2 Causal Discovery

Classical causal discovery methods typically emphasize asymptotic correctness, relying on the assumption of infinite data availability. For example, constraint-based discovery algorithms such as the PC algorithm (Spirtes et al., 2000) and score-based approaches like Greedy Equivalence Search (GES) (Chickering, 2002) provide guarantees of structural recovery in the large-sample limit. Structural equation models (SEMs), including linear non-Gaussian models (Shimizu et al., 2006), similarly offer theoretical identifiability results but often lack explicit finite-sample performance guarantees. Likewise, traditional causal inference methods, such as propensity score matching (Rosenbaum & Rubin, 1983) and instrumental variable techniques (Angrist et al., 1996), generally depend on asymptotic assumptions to ensure valid inference.

Statistical refinements of the aforementioned traditional methods now incorporate finite-sample corrections, enabling explicit sample complexity bounds (Kalisch & Bühlmann, 2007; Wadhwa & Dong, 2021; Yan et al., 2024; Compton et al., 2020). In parallel, modern differentiable and neural causal discovery frameworks, such as NO-TEARS (Zheng et al., 2018), reformulate causal discovery as a continuous optimization problem, achieving improved empirical performance despite typically lacking formal finite-sample guarantees. In the domain of causal inference, doubly robust estimators (Chernozhukov et al., 2018) and conformal inference

methods (Lei & Candes, 2021) explicitly control finite-sample error rates, enhancing the robustness and reliability of inference in practice.

Integrating PAC learning theory with causal methodologies represents a significant recent advancement, enabling formal guarantees on inference accuracy and robustness under resource constraints. While some existing work provides PAC-style results, these efforts are often fragmented, problem-specific, and do not offer a unified theoretical framework for causality. For example, Tadepalli and Russell (Tadepalli & Russell, 2021) proposed a PAC framework for causal trees with latent variables, strategically combining observational data with targeted interventions. Other recent work such as Choo et al. (2024) provides finite-sample PAC guarantees for causal effect estimation via covariate adjustment, including theoretical bounds and algorithms for identifying $\epsilon$–Markov blankets and minimal targeted adjustment sets. Complementary advances have also been made in hypothesis testing for causal Bayesian networks with known structures (Acharya et al., 2018). In contrast, our proposed PACC Analysis framework offers a unified theoretical foundation that generalizes PAC-style reasoning across diverse causal methodologies.

### 3.3 Hypothesis Testing

Building on the foundation of simple hypothesis testing, several extensions are closely related to the proposed PACC framework. One major direction is composite hypothesis testing, where the null and alternative correspond to families of distributions rather than single models Lehmann & Romano (2005). Composite testing naturally arises in modern causal inference problems. For instance, in mediation analysis, the null of "no indirect effect" forms a union of sub-hypotheses, and recent studies have developed valid and powerful tests under this composite structure Huang et al. (2019); He et al. (2024). Similarly, Invariant Causal Prediction (ICP) Peters et al. (2016); Heinze-Deml et al. (2018); Gamella et al. (2020) frames causal discovery as testing whether the conditional distribution $P(Y \mid X)$ remains invariant across environments.

Another important direction is distribution property testing, which emerged directly from the foundations of PAC learning. Decades after Valiant's seminal work Valiant (1984), his sons Gregory and Paul Valiant extended the idea of finite-sample principles to the distribution testing setting Valiant & Valiant (2017), asking whether an unknown distribution possesses a specified property. This paradigm has become increasingly relevant to causal inference, including goodness-of-fit tests Paninski (2008); Chan et al. (2016), kernel-based independence and two-sample tests such as HSIC and MMD Gretton et al. (2008; 2012), and conditional independence tests central to constraint-based causal discovery Cai et al. (2022).

The invariance principle further strengthens the connection between distribution testing and causality. Related approaches exploit nonstationarity or heterogeneity in $P(X)$ or $P(Y \mid X)$ to infer causal directions Zhang et al. (2017); Huang et al. (2020). Anchor regression and its distributional variants Rothenhäusler et al. (2021); Kook et al. (2023) quantify robustness by testing the stability of regression coefficients under perturbations of the data-generating process. More recent developments in high-dimensional conditional independence testing, kernel-based invariance testing, and replicable two-sample tests continue to unify causality with distributional properties Manten et al. (2024); Henzi et al. (2024); Thams et al. (2023).

The proposed PACC Analysis framework can be viewed as a distributional composite hypothesis testing paradigm with PAC-style guarantees. Compared with previous approaches, it generalizes the test from families of distributions to families of causal models. Each causal model $\mathcal{M}$ represents a generative process consistent with causal assumptions such as ignorability or the absence of unmeasured confounding, thereby extending testing from the probability space to the causal space. Moreover, PACC evaluates the worst-case distinguishability between two families of causal models, providing uniform rather than pointwise guarantees. In this setting, one can conceptualize an adversary that selects a specific model pair for the learner to evaluate. The learner must succeed regardless of which pair is chosen, effectively emphasizing the robustness of the algorithm with respect to the target causal concept. In contrast, traditional multiple hypothesis testing often focuses on identifying a single instance where one hypothesis is more easily distinguished from the others (i.e., an "easiest-case" analysis). In summary, PACC Analysis subsumes classical, composite, and distributional hypothesis testing as special cases and provides the first general framework offering uniform, distribution-free, and resource-aware guarantees that quantify the probability of approximately correct causal analysis.

## 4 Applying PACC Analysis to Self-Controlled Case Series

In this section, we demonstrate the efficacy of the PACC Analysis framework in establishing resource-bounded causal discovery guarantees for the SCCS method (Whitaker et al., 2006). The SCCS approach is widely used to identify potential risk factors associated with medical products and has shown strong performance in detecting adverse effects from observational healthcare data (Overhage et al., 2012; Schuemie et al., 2013). However, it is not traditionally considered a causal method. To our knowledge, this is the first work to provide formal causal discovery guarantees for a practical variant of SCCS commonly employed in real-world applications.

### 4.1 SCCS Preliminaries

SCCS was introduced in the 1990s as an innovative method for assessing vaccine safety. It was first used to identify an association between the Measles, Mumps, and Rubella (MMR) vaccine and idiopathic thrombocytopenic purpura (ITP) (Farrington et al., 1995; Farrington, 1995). Since its introduction, SCCS has been widely applied in observational studies, including investigations of the association between antipsychotic drug use and stroke (Douglas & Smeeth, 2008), as well as the risks of acute myocardial infarction and COVID-19 infection (Katsoularis et al., 2021). In 2012, the Observational Medical Outcomes Partnership (OMOP) (Overhage et al., 2012) conducted an empirical evaluation of various statistical methods for adverse drug effect (ADE) detection, finding that SCCS outperformed many alternatives in terms of both predictive accuracy and bias. Subsequent studies have confirmed these findings (Schuemie et al., 2013; Ryan et al., 2013).

The strong empirical performance of SCCS can be intuitively attributed to its self-controlled design, which treats each patient as their own control. This structure inherently adjusts for all time-invariant confounders, including unobserved ones. Below, we introduce the SCCS method and outline the key assumptions on which it relies.

### SCCS Likelihood

Consider a cohort of patients indexed by $i = 1, 2, \ldots, P$, each of whom experiences at least one outcome event. The observation period for each patient is divided into shorter time intervals according to treatment status, indexed by $k$, and time-varying covariates such as age, indexed by $r$. We assume that treatment durations fully cover the period during which the treatment may affect the outcome. For example, in the vaccination study (Miller et al., 1993), the exposure is considered effective for 21 days following vaccine administration.

Let $\phi_i$ denote the baseline event risk for patient $i$, and let $\alpha_{ir}$ represent the effect of the time-varying covariate group $r$. We use $\beta$ to denote the logarithm of the relative incidence rate, capturing the effect of treatment on the outcome event. Define $\beta_{ik}$ as the treatment effect during period $k$, where $\beta_{ik} = \beta$ if patient $i$ is exposed during time period $k$ and $\beta_{ik} = 0$ if unexposed. The number of events $n_{ikr}$ observed for individual $i$ during treatment status interval $k$ and time-varying covariate group $r$ is assumed to follow a Poisson distribution with rate $\lambda_{ikr}$ defined as:

$$\lambda_{ikr} = \exp(\phi_i + \beta_{ik} + \alpha_{ir}), \quad n_{ikr} \sim \text{Poisson}(\lambda_{ikr}\tau_{ikr})$$

where $\tau_{ikr}$ denotes the length of time interval indexed by $k$ and $r$ for patient $i$. The goal of the SCCS method is to estimate the treatment effect parameter $\beta$ by maximizing the conditional log-likelihood:

$$l(\alpha, \beta) = \sum_{ikr} n_{ikr} \log \left( \frac{\tau_{ikr}\exp(\phi_i + \beta_{ik} + \alpha_{ir})}{\sum_{pq} \tau_{ipq}\exp(\phi_i + \beta_{ip} + \alpha_{iq})} \right) \tag{1}$$

A simple example is illustrated in Figure 1. Suppose two individuals are each observed over a 250-day period, with no time-varying covariates involved. Each individual receives a single vaccination during this time, and the exposure effect is assumed to last for 21 days post-vaccination. Then the log-likelihood for these two cases is:

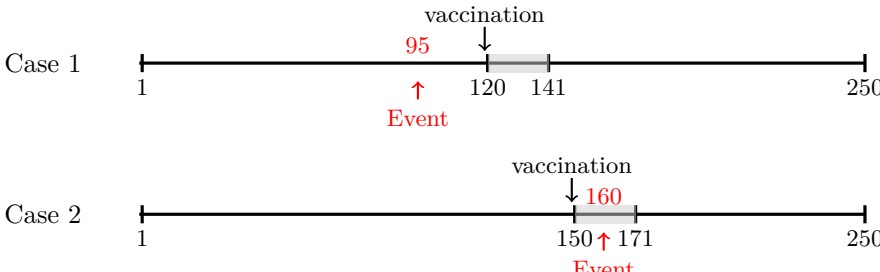

Figure 1: Timelines for two cases illustrating event occurrences and vaccination periods. Red arrows indicate event times, black arrows indicate vaccination dates, and gray shaded boxes represent the post-vaccination risk periods.

$$l = \log\left(\frac{120e^{\phi_1}}{229e^{\phi_1} + 21e^{\phi_1+\beta}}\right)^1 + \log\left(\frac{21e^{\phi_1+\beta}}{229e^{\phi_1} + 21e^{\phi_1+\beta}}\right)^1$$

**Assumptions**

In the potential outcomes framework for causal inference, ignorability (or the "no unmeasured confounders assumption", NUCA) plays a central role. It states that, conditional on observed covariates $X$, the treatment assignment $Z$ is independent of the potential outcome events $Y$. For longitudinal data, this assumption extends to sequential ignorability, which requires that at each time point $t$, the time-varying exposure $Z_t$ is conditionally independent of the potential outcome $Y_t$, given the history of observed covariates $X_t$ and past exposures. The SCCS method, however, offers greater flexibility by allowing for unobserved time-invariant confounders. Instead, it relies on the following weaker condition:

**Assumption 1** (No Unmeasured Time-Varying Confounders (NUTVC)). *Let $Z_t$ denote the exposure assigned at time $t$, and let $\bar{Z}_t = (Z_1, \ldots, Z_t)$ represent the exposure history up to time $t$. Similarly, let $Y_t$ denote the observed outcome at time $t$, with $\bar{Y}_t = (Y_1, \ldots, Y_t)$ representing the outcome history up to time $t$. Let $X_t$ be the vector of observed time-varying covariates at time $t$, and $\bar{X}_t = (X_1, \ldots, X_t)$ the covariate history up to time $t$.*

*We assume that, for all time points $t$, conditional on the entire history of past exposures, outcomes, and time-varying covariates, the exposure at time $t$ is independent of future potential outcomes. Formally,*

$$Z_t \perp Y_{t+1}(\bar{z}_t) \mid \bar{Z}_{t-1}, \bar{Y}_t, \bar{X}_t, \quad \text{for all } t,$$

*where $\bar{z}_t = (z_1, \ldots, z_t)$ is a fixed realization of the exposure history, and $Y_{t+1}(\bar{z}_t)$ is the potential outcome at time $t + 1$ under exposure history $\bar{z}_t$.*

Another common assumption in causal inference is the Stable Unit Treatment Value Assumption (SUTVA) assumption:

**Assumption 2** (SUTVA). *For all units $i$ and time points $t$, the observed outcome $Y_{it}$ depends only on the exposure received by unit $i$, and not on the exposures assigned to any other units $j \neq i$:*

$$Y_{it}(Z_i) = Y_{it}(Z_i, Z_{-i}),$$

*where $Z_i$ is the exposure assigned to unit $i$, and $Z_{-i}$ is the vector of exposures assigned to all other units.*

The following assumption ensures the validity of within-individual comparisons in SCCS. If an outcome event delays or prevents subsequent exposures, it can introduce systematic bias into the estimation of the relative incidence, as observed changes may reflect altered exposure patterns rather than true causal effects. For example, this assumption is violated when the outcome is death, which precludes any future exposures. In such cases, corrective measures such as incorporating pre-exposure risk periods or applying methodological

extensions, are necessary to mitigate bias and preserve the validity of SCCS estimates (Whitaker et al., 2006; Lee & Cheung, 2024).

**Assumption 3.** *Following (Petersen et al., 2016), we adopt the following key assumptions for the SCCS design:*
*(1) The occurrence of an event does not influence the probability of subsequent exposures:*

$$\mathbb{P}(Z_t | \bar{Y}_{t-1}) = \mathbb{P}(Z_t)$$

*(2) Event rates are constant within each defined time interval:*

$$\mathbb{P}(Y_t) = \mathbb{P}(Y_{t'}), \quad \text{for all } t, t' \text{ within the same risk period.}$$

*(3) Events are either independently recurrent or sufficiently rare:*

$$\mathbb{P}(Y_t, Y_{t'}) = \mathbb{P}(Y_t)\mathbb{P}(Y_{t'}) \quad \text{for } t \neq t', \quad \text{or} \quad \mathbb{P}(Y_t) \ll 1.$$

The next assumption aligns with the characteristics of EHRs data and vaccination studies, where the SCCS method is most commonly applied. Given that events are sufficiently rare or independently recurrent, the likelihood of multiple occurrences of the same outcome within a single day is negligible. Moreover, repeated events on the same day are typically interpreted as consequences of an initial occurrence or as recurrences. Therefore, it is reasonable to assume that the number of daily events follows a binomial distribution, which approximates a Poisson distribution as the observation period increases and the event incidence remains low.

**Assumption 4** (Single Event per Day). *Only the first occurrence of a specific outcome type within a single day is recorded.*

Under Assumptions 1–4, the SCCS method provides consistent estimates of the relative incidence associated with exposure (Farrington, 1995; Whitaker et al., 2006). A significant deviation of the estimated parameter $\beta$ from 0, or equivalently of $\exp(\beta)$ from 1, indicates a causal effect of the treatment on event incidence. In addition, we require that the probability of exposure $Z$, denoted by $\lambda := \mathbb{P}(Z = 1)$, is sufficiently large to ensure a minimal fraction of exposure time across the population, independent of $Y$. This condition is formalized in the following assumption:

**Assumption 5** (Active-mass lower bound). *There exists $\lambda_{\min} \in (0, 1]$ such that the per-patient active (exposure) mass satisfies $\lambda \geq \lambda_{\min} > 0$.*

### 4.2 SCCS Under the PACC Analysis Framework

We define the causal concept $c$ for SCCS as whether exposure $Z$ influences the incidence of experiencing the event $Y$. Specifically, we require the change in event probability to be sufficiently large such that $\frac{\mathbb{P}(Y|Z)}{\mathbb{P}(Y|\neg Z)} \geq \delta$, for some $\delta > 0$, where $\neg Z$ denotes the absence of exposure. We refer to this condition as "$Y$ is $\delta$-dependent on $Z$." The corresponding causal family is defined as $\mathcal{F}_{\delta,c} = \{\langle \mathcal{M}_{j1}, \mathcal{M}_{j2} \rangle\}$. In $\mathcal{M}_{j1}$, $Y$ follows a Poisson distribution with intensity that depends only on a time-invariant, patient-specific latent variable $\phi_i$. In $\mathcal{M}_{j2}$, the intensity of $Y$ is a function of both $\phi_i$ and the exposure status $Z$, with the effect of $Z$ captured by the coefficient $\beta$.

**Theorem 1.** *Under Assumptions 1-5, for any $0 < \delta$ and $0 < \epsilon < 1$, we define the target causal concept $c$ as whether the event $Y$ is $\delta$-dependent on the exposure $Z$ within the instance space $\mathcal{I}$. The corresponding causal family is given by $\mathcal{F}_{\delta,c} = \{\langle \mathcal{M}_{j1}, \mathcal{M}_{j2} \rangle\}_{j \geq 1}$ as defined above.*

*Then, given $\mathcal{O}\left(\frac{1}{\lambda_{\min}^2 \log^2(\delta)} \log\left(\frac{1}{\epsilon}\right)\right)$ test examples, the SCCS algorithm can correctly distinguish between $\mathcal{M}_{j1}$ and $\mathcal{M}_{j2}$ with probability at least $1 - \epsilon$. Therefore, $c$ is PACC discoverable by the SCCS method.*

The proof of the theorem is provided in Appendix A.1. The algorithmic formulation and implementation details are presented in Algorithm 2. We also include a corollary in Appendix A.2 showing that, under the same conditions but without Assumption 5, if $\lambda$ is not bounded away from zero by $\lambda_{\min}$, the target remains identifiable but is not PACC-discoverable uniformly over the model class. This result helps illustrate the connection between our PACC framework and the classical causal identifiability argument.

---

**Algorithm 2** PACC Discoverability with SCCS

---

**Requirements** Parameters: $\epsilon \in (0,1)$, $\delta > 0$. Target causal concept $c$: whether event $Y$ is $\delta$-dependent on exposure $Z$. Causal family $\mathcal{F}_{\delta,c} = \{\langle \mathcal{M}_{j1}, \mathcal{M}_{j2} \rangle\}_{j \geq 1}$, where:

- Under $\mathcal{M}_{j1}$: $Y$ is $\delta$-dependent on $Z$.

- Under $\mathcal{M}_{j2}$: $Y$ is independent of $Z$.

1: **Input:** $\epsilon, \delta$, any model pair $\langle \mathcal{M}_{j1}, \mathcal{M}_{j2} \rangle \in \mathcal{F}_{\delta,c}$, and a sample $S$ generated from one of the two models with size $\mathcal{O}\left(\frac{1}{\lambda_{\min}^2 \log^2(\delta)} \log\left(\frac{1}{\epsilon}\right)\right)$.

2: **Test:** Estimate the log-relative incidence $\beta$ using SCCS from sample $S$ via Equation 1. Decide whether $Y$ is independent of $Z$ based on the following criterion:

3: **if** $\beta \geq \log(\delta)/2$ **then**

4:     **output:** $\mathcal{M}_{j1}$ is identified as the data-generating model.

5: **else**

6:     **output:** $\mathcal{M}_{j2}$ is identified as the data-generating model.

7: **end if**

8: **Guarantee:** The SCCS algorithm correctly distinguishes $\mathcal{M}_{j1}$ from $\mathcal{M}_{j2}$ with probability at least $1 - \epsilon$.

---

One potential extension involves addressing the current limitation that observed time-varying covariates are excluded. While such covariates can often be adjusted for using weighting or matching techniques, a more general SCCS framework may incorporate them directly, such as through $\alpha_{ir}$ in Equation 1. In Theorem 1 and Algorithm 2, we consider the simplified setting where $r = 1$, meaning no time-varying covariates are included. Extending the PACC framework to the case with time-varying covariates is a natural next step. In such cases, the likelihood becomes a function of both the drug effect $\beta$ and an additional nuisance parameter $\alpha$ representing the time-varying effect. The inclusion of these nuisance parameters may reduce the statistical efficiency of estimating $\beta$, thereby increasing the required sample size.

In addition, several notable extensions of the SCCS framework have been proposed. Simpson and Madigan (Simpson et al., 2013) introduced the Multiple SCCS (MSCCS) model, which enables the simultaneous analysis of multiple drugs rather than a single intervention. Kuang (Kuang et al., 2017) further extended SCCS to incorporate time-varying, patient-specific baseline risks, partially addressing the challenge of unmeasured time-varying confounders. In such cases, the definition of the causal concept and the design of competing causal models would require more deliberate and context-specific considerations. However, these directions go beyond the scope of this foundational work, and a detailed analysis of these extensions is left for future research.

## 5 Analyzing Other Algorithms Using the PACC Analysis Framework

In this section, we demonstrate the alignment between the PACC Analysis framework and classical causal literature, including results from both graphical causal models and the potential outcomes framework. This alignment illustrates that PACC Analysis can express and extend existing results by translating traditional asymptotic inference into a PAC-style analysis, thereby shedding new light on the utility of these methods under finite-sample constraints. Importantly, we emphasize that the assumptions commonly imposed to ensure the validity of classical methodologies often coincide with those required to derive favorable theoretical guarantees in the PACC framework.

### 5.1 Propensity Score

In observational studies, covariate imbalance and confounding variables pose significant challenges to causal inference. To mitigate these challenges, researchers often use propensity score methods (Rosenbaum & Rubin, 1983), which estimate the probability of receiving a specific treatment, typically via logistic regression. Techniques such as matching, weighting, or stratification based on these scores aim to replicate the balance

achieved in RCTs, thereby supporting valid causal conclusions. This section extends the PACC framework by integrating it with the foundational ideas of propensity scoring.

We introduce the following notation. The instance space $\mathcal{I}$ consists of feature vectors defined over $n$ variables, denoted as $\{X, Y, Z\}$, where $Z \in \{0, 1\}$ represents treatment assignment, $X$ includes all pre-treatment covariates, and $Y \in \{0, 1\}$ is the outcome event. A specific configuration of the covariates is written as $X = x$, and similarly for $Y$ and $Z$. For each individual $i$, we define the potential outcomes $(Y_i(0), Y_i(1))$, corresponding to the outcomes under treatment conditions $Z = 0$ and $Z = 1$ respectively. The average treatment effect (ATE) is then estimated as:

$$\text{ATE} = \mathbb{E}_X[Y|Z = 1] - \mathbb{E}_X[Y|Z = 0]. \tag{2}$$

Feature vectors in the instance space $\mathcal{I}$ are drawn from a joint probability distribution $\mathcal{D}(x, y, z)$, which can be factorized as a product of three components: $\mathcal{P}$, $\mathcal{Q}$, and $\mathcal{R}$:

$$\mathcal{D}(x, y, z) = \mathcal{Q}(X = x)\mathcal{P}(Z = z|X = x)\mathcal{R}(Y = y|Z = z, X = x).$$

In this setup, $\mathcal{Q}$ models the distribution of patient covariates and is well-behaved [2]. $\mathcal{P}$ captures how treatment assignment is determined based on covariates in observational studies, while $\mathcal{R}$ models subsequent outcome events, conditional on treatment and covariates. In a randomized experiment, $\mathcal{Q}$ still describes the target population, but the treatment assignment follows a Bernoulli distribution with equal probability (i.e. $\mathcal{B}_{0.5}$). As a result, the treated sample follows the distribution $\mathcal{Q}\mathcal{B}_{0.5}$ rather than $\mathcal{Q}\mathcal{P}$ as in the observational settings. The primary goal of propensity scoring is to use observational data drawn from $\mathcal{Q}\mathcal{P}\mathcal{R}$ to emulate the RCT distribution $\mathcal{Q}\mathcal{B}_{0.5}\mathcal{R}$.

We impose the following standard assumptions, commonly used in propensity score-based analyses (Rosenbaum & Rubin, 1983; 1984; Dehejia & Wahba, 2002):

**Assumption 6** (Ignorability). $(Y(0), Y(1)) \perp Z|X$.

**Assumption 7** (Consistency). $Y = Y(Z)$.

**Assumption 8** (Bounded Positivity). *For $0 < \delta_1 < 1$, $\delta_1 < \mathcal{P}(Z = 1|x) < 1 - \delta_1$, $\forall x \in X$.*

**Assumption 9** (Logistic Model for Treatment). *The treatment assignment $\mathcal{P}$ can be represented by a logistic regression model over $X$.*

We assume that the treatment has a non-negligible effect, denoted as $\delta_2 = \mathbb{P}(Y|Z) - \mathbb{P}(Y|\neg Z) \in (0, 1)$. Let $\delta = \min\{\delta_1, \delta_2\}$, and characterize the causal relationship by assessing whether the outcome $Y$ is $\delta$-dependent on $Z$. Under these assumptions, we present Theorem 2 and the corresponding Algorithm 3, along with an outline of the proof; full details are provided in Appendix B.1.

**Theorem 2.** *Under Assumptions 6-9, let $\mathcal{I}$ be the instance space consisting of $n$ variables $\{X, Y, Z\}$ as defined above. For any $0 < \delta < 1$ and $0 < \epsilon < 1$, define the target causal concept $c$ as whether $Y$ is $\delta$-dependent on $Z$, with the corresponding causal family given by $\mathcal{F}_{\delta,c} = \{\langle \mathcal{M}_{j1}, \mathcal{M}_{j2} \rangle\}_{1 \leq j}$.*

*Let $\gamma = \min\{\epsilon, \delta, \frac{\delta^2}{4}\}$. Then given $\mathcal{O}\left(\frac{1}{\gamma^3} \log(\frac{1}{\gamma})\right)$ test examples, the propensity score method can correctly distinguish between any pair $\langle \mathcal{M}_{j1}, \mathcal{M}_{j2} \rangle$ with probability at least $1 - \epsilon$. Therefore, $c$ is PACC discoverable by propensity score method.*

Our goal is to approximate the randomized distribution $\mathcal{Q}\mathcal{B}_{0.5}\mathcal{R}$ from the observational distribution $\mathcal{Q}\mathcal{P}\mathcal{R}$. We start by developing a linear propensity model $\mathcal{P}'$ to approximate $\mathcal{P}$, using logistic regression on pre-treatment variables. The sample complexity required for this approximation is polynomial, as established by Kearns and Schapire in their foundational work on learning probabilistic concepts (p-concepts) (Kearns & Schapire, 1994), and by Haussler's extension of VC dimension theory to p-concepts via pseudodimension (Haussler, 1992). Because the pseudodimension of linear models is low, $\mathcal{P}'$ can approximate $\mathcal{P}$ with high accuracy and confidence. This approximation enables rejection sampling based on the estimated propensity

---

[2]$\mathcal{Q}$ should satisfy certain measurability conditions, including bounded support, absolute continuity, and Lipschitz continuity. More details can be found in (Van der Vaart & Wellner, 2000)

---

**Algorithm 3** PACC Discoverability Using Propensity Score

---

**Requirements** Parameters: $\epsilon \in (0,1), \delta \in (0,1)$. Instance space $\mathcal{I}$ contains $n$ variables denote as $\{X, Y, Z\}$. Target causal concept $c$ and the corresponding causal family $\mathcal{F}_{\delta,c} = \{\langle \mathcal{M}_{j1}, \mathcal{M}_{j2} \rangle\}_{1 \leq j}$, where:

- Under $\mathcal{M}_{j1}$, $Y$ is $\delta$-dependent on $Z$.

- Under $\mathcal{M}_{j2}$, $Y$ is independent of $Z$.

**Input:** A sample $S$ of size $\mathcal{O}\left(\frac{1}{\gamma^3} \log(\frac{1}{\gamma})\right)$ drawn from $\mathcal{M}_{j1}$ or $\mathcal{M}_{j2}$, where $\gamma = \min\{\epsilon, \delta, \frac{\delta^2}{4}\}$.

**Step 1: Propensity Model Estimation.** Build a propensity model $\mathcal{P}'$ that approximates $\mathcal{P}$ using $\frac{64}{\gamma^2}(2n \log(\frac{16e}{\gamma}) + \log(\frac{48}{\epsilon}))$ samples to predict treatment assignment.

**Step 2: Rejection Sampling.** Obtain $\mathcal{O}\left(\frac{1}{\gamma^2} \log(\frac{1}{\epsilon})\right)$ samples from $\mathcal{Q}\frac{\mathcal{P}}{\mathcal{P}'}$ by rejection sampling.

**Step 3: Final Decision.** Compute the ATE using Equation (2) on the adjusted sample:
**if** ATE $\geq \delta/2$ **then**
    **output:** $\mathcal{M}_{j1}$ is identified as the data-generating model.
**else**
    **output:** $\mathcal{M}_{j2}$ is identified as the data-generating model.
**end if**
**Guarantee:** The algorithm correctly distinguishes between $\mathcal{M}_{j1}$ and $\mathcal{M}_{j2}$ with probability at least $1 - \epsilon$.

---

distribution $\mathcal{P}'$, thereby generating a new sample from $\mathcal{Q}\frac{\mathcal{P}}{\mathcal{P}'}$, which closely mimics the randomized treatment assignment of $\mathcal{QB}_{0.5}$. To determine the total number of samples required, we use results from agnostic PAC learning (Kearns et al., 1992), accounting for both the data needed to fit the propensity model and the rejection sampling process. Finally, we determine that $Y$ is independent on $Z$ if the ATE is smaller than $\frac{\delta}{2}$.

We note a potential suboptimality in the sample complexity stated in Theorem 2 compared with that of the uniformly most powerful likelihood ratio test. This difference arises not from analytical looseness, but from the compounded dependence introduced by the three-stage structure of the proof. Future research may investigate whether alternative techniques, such as overlap weighting or propensity score matching, could improve sample efficiency or broaden the framework's applicability.

## 5.2 Instrumental Variable

This subsection explores another predominant method in observational studies for addressing confounders: instrumental variables (IVs). Introduced in the early 20th century (Wright, 1928), IV methodology has fostered a dynamic and fruitful area of research, with significant applications in the fields of economics, statistics, and medicine (Sexton & Hebel, 1984; Card, 1993; Angrist & Pischke, 2008). The intuition behind IVs is to identify an external variable, an instrument, that influences the treatment but has no direct effect on the outcome except through the treatment. This setup allows researchers to isolate the exogenous component of the treatment variation that causally impacts the outcome.

While the propensity score results rely on the ignorability assumption, this condition is often violated in practice due to the unmeasured confounders. To address this limitation, we explore how IVs provide a potential solution by relaxing Assumption 6 and replacing it with the following assumption:

**Assumption 10** (Perfect IV). *Let $D$ be a perfect IV, with the causal models follow the structure illustrated in Figure 2. The variable $D$ satisfies the following three conditions:*

- *(Relevance) $D$ is correlated with the treatment $Z$: $D \not\perp\!\!\!\perp Z | X$ ;*

- *(Independence) $D$ is conditionally independent of all unmeasured confounders $U$ that affect the outcome $Y$: $D \perp\!\!\!\perp U | X$;*

- *(Exclusion restriction) $D$ influences the outcome $Y$ only through its effect on the treatment $Z$: $Y(D, Z) = Y(Z)$.*

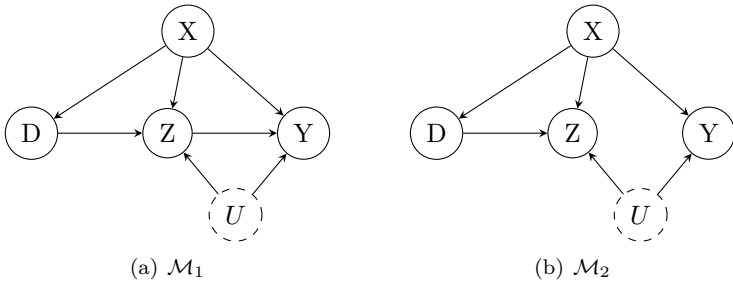

(a) $\mathcal{M}_1$          (b) $\mathcal{M}_2$

Figure 2: Bayesian network structures that define the relationships between the outcome $Y$, treatment $Z$, unmeasured confounders $U$, an instrumental variable $D$, and all other observed covariates $X$.

---

**Algorithm 4** PACC Discoverability Using 2SLS

---

**Requirements** Parameters: $\epsilon \in (0,1), \delta \in (0,1)$. Instance space $\mathcal{I}$ contains $n$ variables denoted as $\{Z, Y, X, U, D\}$. Target causal concept $c$ and the corresponding causal family $\mathcal{F}_{\delta,c} = \{\langle \mathcal{M}_{j1}, \mathcal{M}_{j2}\rangle\}_{1\leq j}$, where:

- Under $\mathcal{M}_{j1}$, $Y$ is $\delta$-dependent on $Z$.

- Under $\mathcal{M}_{j2}$, $Y$ is independent of $Z$.

**Input:** Sample $S$ of size $\mathcal{O}\left(\frac{1}{\delta^2 \epsilon}\right)$, drawn from either $\mathcal{M}_{j1}$ or $\mathcal{M}_{j2}$.

**Stage I: Regress $Z$ on $D$:** $Z = \alpha D + \xi_1$, $\hat{\alpha} = \frac{\sum_i D_i Z_i}{\sum_i D_i^2}$, $\hat{Z} = \hat{\alpha} D$

**Stage II: Regress $Y$ on $\hat{Z}$:** $Y = \beta \hat{Z} + \xi_2$, $\hat{\beta} = \frac{\sum_i D_i Y_i}{\sum_i D_i Z_i}$

**Final Decision**
**if $\hat{\beta} \geq \delta/2$ then**
    **output:** $\mathcal{M}_{j1}$ is identified as the data-generating model.
**else**
    **output:** $\mathcal{M}_{j2}$ is identified as the data-generating model.
**end if**
**Guarantee:** The 2SLS method correctly distinguishes between $\mathcal{M}_{j1}$ and $\mathcal{M}_{j2}$ with probability at least $1 - \epsilon$.

---

Under Assumption 10, we apply d-separation to the full causal DAGs $\mathcal{M}_1$ and $\mathcal{M}_2$ (Figure 2) to derive reduced models over the observed variables $(X, D, Y)$ (Figure 3). In $\mathcal{M}_1$, the path $D \to Z \to Y$ induces a dependence between $D$ and $Y$, which remains active as long as $Z$ is not conditioned on. After marginalizing over the unobserved confounder $U$, this dependence is preserved and represented as an effective direct edge $D \to Y$ in the simplified graph $\mathcal{M}_1'$, indicating that $D \not\perp\!\!\!\perp Y \mid X$. In contrast, in $\mathcal{M}_2$, once we condition on $X$, all paths between $D$ and $Y$ are blocked. This follows from two structural properties: (i) the exclusion restriction ensures there is no direct edge from $D$ to $Y$, and (ii) conditioning on $X$ blocks any back-door paths through the unobserved confounder $U$. After marginalizing over $U$, the resulting simplified graph $\mathcal{M}_2'$ contains no direct edge between $D$ and $Y$; instead, both share a common parent $X$, forming a v-structure $D \leftarrow X \to Y$. In this configuration, conditioning on $X$ d-separates $D$ and $Y$, implying $D \perp\!\!\!\perp Y \mid X$.

With the simplified Bayesian network, we implement the just-identified case of the Two-Stage Least Squares (2SLS) algorithm and demonstrate its applicability within the PACC framework. The 2SLS method is widely used in econometrics and causal inference, particularly when randomized experiments are not feasible and unmeasured confounding threatens the validity of observational analyses. The just-identified 2SLS procedure proceeds in two stages:

**Stage I:** Regress the endogenous variable $Z$ on the instrument $D$ to obtain predicted values, i.e.: $Z = \alpha D + \xi_1$;
**Stage II:** Regress the outcome $Y$ on the predicted values $\hat{Z}$ obtained from the first stage, i.e., $Y = \beta \hat{Z} + \xi_2$.

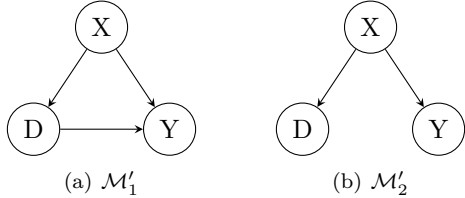

(a) $\mathcal{M}'_1$        (b) $\mathcal{M}'_2$

Figure 3: Simplified Bayesian network structures specify the relationships between the outcome $Y$, instrumental variable $D$, and all other measured variables $X$.

Here, $\xi_1$ and $\xi_2$ are error terms with mean zero and finite variance. Similar to the propensity score analysis, we define the target causal concept $c$ as whether the outcome $Y$ is $\delta$-dependent on the treatment $Z$, where $\delta = \min\{\delta_1, \delta_2\}$. Here, $\delta_1 \in (0, 1)$ comes from the positivity assumption, and $\delta_2 \in (0, 1)$ denotes the minimum difference between the probabilities of $Y$ occurring when $Z$ is true versus when $Z$ is false, specifically among compliers—individuals whose treatment status changes in response to the instrument. This quantity captures the minimal causal effect of the instrument on the outcome, mediated through the treatment, within the subpopulation affected by the instrument. Under these assumptions, we present Theorem 3 and the corresponding Algorithm 4.

**Theorem 3.** *Under Assumptions 7-8, let $\mathcal{I}$ be the instance space consisting of the binary variables: treatment $Z$, outcome $Y$, measured covariates $X$, unmeasured confounders $U$, and a perfect IV $D$ that together satisfy Assumption 10.*

*For any $0 < \delta < 1$ and $0 < \epsilon < 1$, define the target causal concept $c$ as whether $Y$ is $\delta$-dependent on $Z$, with the corresponding causal family given by $\mathcal{F}_{\delta,c} = \{\langle \mathcal{M}_{j1}, \mathcal{M}_{j2}\rangle\}_{1 \leq j}$. Then given a test sample of size $\mathcal{O}\left(\frac{1}{\delta^2\epsilon}\right)$, the 2SLS method can correctly distinguish between any pair $\langle \mathcal{M}_{j1}, \mathcal{M}_{j2}\rangle$ with probability at least $1 - \epsilon$. Therefore, $c$ is PACC discoverable via 2SLS methods.*

To prove the theorem, we bound both the false positive and false negative errors. Specifically, when the true treatment effect $\beta = 0$ (i.e., no causal effect), the estimated coefficient $\hat{\beta}_{2\mathrm{SLS}}$ should, with high probability, satisfy $|\hat{\beta}_{2\mathrm{SLS}}| \leq \delta/2$. Conversely, when the true effect satisfies $|\beta| \geq \delta$, the estimator should exceed the decision threshold, i.e., $|\hat{\beta}_{2\mathrm{SLS}}| > \delta/2$, with high probability. These guarantees are established by applying Chebyshev's inequality under the assumption that the covariance between $Z$ and $D$, and the variance of $D$, are finite. The complete proof is provided in Appendix B.2.

# 6 Conclusions and Future Work

The first contribution of this paper is to highlight the need for causal models tailored to resource-limited settings, especially in observational studies. The proposed PACC Analysis framework offers new theoretical insights that inspire novel algorithms. It also improves our understanding of existing methods by quantifying their resource requirements. The third contribution is to present some initial results within the PACC Analysis framework, demonstrating its practical value and potential. Specifically, we provide the first theoretical guarantee for the SCCS method in causal discovery. We also offer theoretical justifications for established methods such as propensity score and instrumental variables.

Future research can extend in several directions. One direction is to refine the current PACC Analysis framework. Although it is designed to be broadly applicable, this paper focuses on the causal concept of whether an outcome depends on a treatment. Future work could explore alternative concepts, such as causal direction (we include an example of this in Appendix C), where competing models imply opposite causal directions, and causal structure, where model pairs differ in their structural assumptions. Another open question concerns the number of causal model pairs. While the total number of pairs can be large, practical constraints and background knowledge often reduce the number of relevant comparisons. A promising direction for future work is to establish theoretical bounds that relate the number of model pairs to the required sample size.

Additionally, exploring variations of the PACC Analysis framework opens new avenues for research. One promising direction is a Bayesian variant of PACC Analysis. This approach would begin with a prior distribution over all plausible causal models and aim to either identify the correct model or converge to an accurate posterior distribution with high probability. Such an extension would parallel the role of U-Learnability (Muggleton & Page, 1998), which redefined PAC-learnability by introducing a probability distribution over target concepts in addition to the distribution over training examples. However, this direction poses significant computational challenges: computing exact posteriors over DAGs is known to be intractable, and approximate methods such as MCMC-based sampling often struggle to scale to large model spaces.

A further direction is to extend the PACC Analysis framework to a broader class of causal algorithms. For example, under appropriate assumptions, the framework can be applied to differential prediction tasks, which aim to identify models that perform better for one subgroup than another. For instance, predicting postpartum depression more accurately in younger women than in older women, or predicting preeclampsia more effectively for one racial group than another (Kuusisto et al., 2014). The PACC Analysis framework provides a formal, sample-efficient lens to interpret such disparities causally. In particular, it enables principled testing of whether a sensitive attribute has a causal influence on predictions or outcomes. In this spirit, PACC Analysis supports a "probably approximately fair" paradigm, wherein fairness properties hold with high probability and small error under limited data, paralleling how PACC ensures causal correctness. Given its flexibility and resource-aware design, the PACC Analysis framework holds promise as a theoretical foundation for deriving approximate learning guarantees and sample complexity bounds across a wide range of causal algorithms.

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
