# OpenReview forum: "Probably Approximately Correct Causal Discovery"
_TMLR — Rejected by TMLR_

### Review · Reviewer_jqms · 2025-09-17

**Summary Of Contributions:**

This paper proposes the Probably Approximately Correct Causal (PACC) Discovery framework, which adapts PAC learning theory to the domain of causal discovery. The key motivation is to provide finite-sample theoretical guarantees for causal inference methods, so that one can argue that certain causal concepts can be recovered with high probability ($\geq 1 - \delta$) and bounded error ($\leq \varepsilon$), using sample sizes that are polynomial in $1/\varepsilon$ and $1/\delta$.

The authors instantiated this framework for three causal concepts:

(1) Self-Controlled Case Series (SCCS), in Section 4

(2) Propensity score, in Section 5.1

(3) Instrumental variable, in Section 5.2

**Additional Comments:**

I feel that this work is not *really* about PAC learning but hypothesis testing since the proposed method only compares and chooses between a pair of concepts.

It would also be nice if the authors provided some kind of hardness/lower bound results for sample complexity so that readers can appreciate the sample efficiency of their proposed method.

**Audience:**

Yes

**Audience Explanation:**

There is a lack of finite-sample results in causal inference, so this would be of interest to the causal inference community.

**Broader Impact Concerns:**

Nil

**Claims And Evidence:**

No

**Claims Explanation:**

Proofs were given in the appendix but I have quite a bit of clarifications and questions. Please see "Requested changes" section below.

**Requested Changes:**

## General comment
Can you be explicit about where each specific assumption is being invoked or used (especially in the proofs)? It is currently unclear if and why each assumption is required for your method to work.

## Clarifications and questions
- Definition 4: What happens if the samples are coming neither from $M_{j1}$ nor $M_{j2}$?
- In standard causal discovery, we want to output one Bayesian network or at least a family (e.g. Markov equivalence class). In that case, does your method require exponentially many pair of comparisons?
- Last sentence of Page 9: "we require that the probability of exposure $Z$... without dependence on $Y$". Is this a required assumption for the method to work? If so, please make it explicit.
- Theorem 1: I think that the big-O notation should not hide the $1/\lambda^2$ factor. Also, the proof uses Hoeffding to argue that $|\hat{\beta} - \mathbb{E}[\hat{\beta}]| < \frac{\log \delta}{2}$ given sufficient samples, where $\beta$ is the MLE given observations. Why does this imply the claim in Theorem 1? There seems to be a leap in argument.
- First paragraph after Algorithm 2 environment: it's a bit strange that the paper defines $\alpha_{ir}$ on Page 8 but then never uses it
- Theorem 2 and Algorithm 3: Is $\frac{\mathcal{P}}{\mathcal{P}'} = \mathcal{B}_{0.5}$? Also, shouldn't the sample complexity depend on $\mathcal{P}'$?
- Section 5.2 and Algorithm 4: The algorithm seems to be assuming particular functional forms of the graphical networks in Figure 2. Can you explicitly write out the corresponding structural equation models (SEMs) to be precise and clear?
- Top of page 2 of appendix: Why is this $\hat{\beta}$ the right estimate? Can you show the derivation or point to a reference?
- Middle of page 2 of appendix: Can you give a proof or reference the inequality $|\log(S_{\ell}) - \log(\mathbb{E}[S_{\ell}])| \leq ...$?
- Step 2 on page 4 of appendix: Why is $p_{accept}$ defined as such? Where is $\mathcal{P}$? Also, how is $p_{avg}$ formally defined? Is $p_{avg} = p_{accept}$?
- Equation (4) in appendix: Can you show the derivation for this?

## Typos and writing suggestions
- Page 2: "enrichs" should be "enriches"
- Extra space between full stop and the first footnote label
- Third sentence in first paragraph after Algorithm 2 environment: "excludes" should be "excluded"
- First sentence in second paragraph after Algorithm 2 environment: "n addition" shoudl be "In addition"
- Equation environment after equation (2): Write $\mathcal{D}(x,y,z)$ instead of just $\mathcal{D}$ to emphasize dependence on the parameters?
- Paragraph right after: "In a randomized experiment, the $\mathcal{Q}$ still describes..." should be just "In a randomized experiment, $\mathcal{Q}$ still describes..."
- Theorem 2 statement: Missing/extra bracket in big-O term
- 4th line of equation from the bottom of page 2 of the appendix: Missing closing brackets of the expectation terms
- Top of page 4 of appendix: it is weird to say $\delta' = \delta - \varepsilon$. Instead, say that $\delta - \varepsilon \leq \delta'$ implies $\mathbb{E}_{X \sim \mathcal{Q}\mathcal{P}'}[f(X)] = \frac{1}{\delta'} \int_X \ldots dx \leq \frac{1}{\delta - \varepsilon} \int_X \ldots dx \leq (\text{your final term})$

---

> ### Author Response · Authors · 2025-10-18
>
> We sincerely thank the reviewer for their careful reading, detailed comments, and constructive suggestions.
> Below, we address each point in detail. We have added clarifying sentences, corrected typos, and updated theorems and derivations in the revised version.
>
> ### **1. PACC Framework Definition**
>
> **(a) What happens if the samples are coming neither from $\mathcal{M} _ {j1}$ nor $\mathcal{M} _ {j2}$?**
>
> Thank you for raising this important point.  In the classical PAC learning framework, if no samples are labeled according to the target concept, the learning problem becomes ill-posed, and there is no requirement for the learning algorithm to succeed.  Analogously, in the PACC Discovery framework, if the observed samples are generated from neither $\mathcal{M} _ {j1}$ nor $\mathcal{M} _ {j2}$, this implies that we do not have access to the true data-generating process corresponding to either candidate model. In such cases, the learner is not expected to succeed, and no performance guarantees are required.
>
> **(b) Does your method require exponentially many pairs of comparisons?**
>
> We appreciate this insightful question. In the worst case, the number of model pairs in the causal family $\mathcal{F} _ {\delta,c}$ could indeed grow exponentially if no structural or domain constraints are imposed. However, our framework does not require enumerating or evaluating all possible pairs. The key idea of PACC-discoverability is that the learner is required to succeed on any pair of models that an adversary may choose. This follows the standard PAC philosophy: the guarantee holds uniformly over the family, but the learning procedure itself only needs to produce a hypothesis that distinguishes one arbitrary but fixed pair with the desired $(\epsilon,\delta)$-level reliability.
>
> Moreover, in practical applications, we explicitly encourage restricting the comparison set to a smaller and interpretable subset of competitors. This can be achieved through several strategies: (i) incorporating background knowledge, such as fixing certain known edges or non-edges based on prior causal information; (ii) bounding model complexity, for example by limiting the number of variables, the size of separating sets, or the maximum number of parents per node; and (iii) focusing on local edge-level tests, as demonstrated in our examples, where each comparison is restricted to models differing by a single edge (e.g., $Z \rightarrow Y$).
>
> ### **2. SCCS Part**
>
> We thank the reviewer for the helpful observation regarding the SCCS section.  As a brief summary, we have made the following revisions and clarifications:
>
> (a) Changed the last sentence on Page 9 into a formal assumption specifying the minimum mass parameter $\lambda \ge \lambda_{\min}$.
>
> (b) Incorporated $\lambda_{\min}$ into the big-$O$ notation appearing in both Theorem 1 and Algorithm 2.
>
> (c) Added a new corollary stating that if $\lambda$ is not bounded away from zero, the causal target remains identifiable but is not PACC-discoverable.
>
> (d) Expanded the explanation of why the proof supports the claim in Theorem 1 and clarified why $\hat{\beta}$ is the appropriate estimator.
>
> (e) Introduced a lemma that provides a detailed proof of the inequality in the middle of Page 2.
>
> (f) Added a brief explanation for the definition of $\alpha_{ir}$ and its role, noting why it does not appear explicitly in the subsequent derivation.
>
> **(a) \& (b) New assumption and updated theorem about $\lambda$**
>
> Great questions, and thank you for flagging them! You are right that the $\lambda$-dependence should be included in the big-$O$ term. Below, we modify the last line of Page 9 by introducing a new assumption to formalize $\lambda_{\min}$ and to clarify where it is used in the theorem and proof. As in the paper, let $Z$ denote the exposure, and define the exposure mass parameter as $\lambda := \Pr(Z=1)$ or $\lambda := \mathbb{E}\left[\frac{\iota_{\mathrm{trt}}}{\iota_{\mathrm{all}}}\right]$. Then we have the following assumption:
>
> > **Assumption (Active-mass lower bound).**
> >There exists $\lambda _ {\min}\in(0,1]$ such that the per-patient active/event mass satisfies $\lambda \ge \lambda_{\min} > 0$.
>
> The lower bound $\lambda _ {\min}$ serves as a population-level constant ensuring a minimal fraction of exposure time across all individuals. Under this assumption, the proof remains valid by replacing $\lambda$ with $\lambda_{\min}$. Accordingly, we have updated Theorem 1 and Algorithm 2 to include $\lambda_{\min}$ in the required sample size, which now scales as
> $
> \mathcal{O}\left(\frac{1}{\lambda _ {\min}^{2}\log^{2}(\delta)}\log\left(\frac{1}{\epsilon}\right)\right).
> $

---

> ### Author Response · Authors · 2025-10-19
>
> **(c) New corollary when $\lambda$ is not bounded**
>
> As an extension of the $\lambda _ {\min}$ assumption, we have added a new corollary in **Appendix A.2**, showing that if $\lambda$ is not bounded away from zero by $\lambda _ {\min}$, the target concept remains identifiable but is no longer PACC-discoverable. This result helps clarify the connection between our PACC framework and the classical notion of causal identifiability.
>
> **(d) $\hat{\beta}$ in Theorem 1**
>
> We use the standard conditional likelihood formulation in Equation (1) (see section 3.2 of [1] or section 2.1 of [2]). Differentiating and setting $\partial l/\partial\beta = 0$ yields
> $
> \hat{\beta}
> = \log S_{1}-\log S_{2}.
> $
> Thus, $\hat{\beta}$ is the maximum-likelihood estimator (MLE) of the log-rate ratio between the exposed and unexposed windows.
>
> To connect this estimator with Theorem1, note that under the true model $\mathcal{M} _ {j1}$ with causal risk ratio
> $\delta = e^{\beta} > 1$, we have $\mathbb{E}[\hat{\beta}] = \log \delta > 0$, whereas under the null model $\mathcal{M} _ {j2}$
> (no effect), $\mathbb{E}[\hat{\beta}] = 0$.
> Define the concentration event
> $
> \mathcal{E} := \bigcap _ {\ell=1}^2 \big(\{|\log S_\ell - \log \mathbb{E}[S_\ell]| < \frac{1}{4}\log\delta\} \big).
> $
> Using Hoeffding’s inequality together with the Log–Lipschitz reduction (Lemma 1), we obtain
> $
> \mathbb{P}(\mathcal{E}^\complement)
> \le
> 4\exp\Big(-\frac{P\lambda_{\min}^2\log^2\delta}{8}\Big).
> $
> Thus, choosing
> $
> P
> \ge
> \frac{8}{\lambda_{\min}^2\log^2\delta}
> \log\Big(\frac{4}{\epsilon}\Big)
> $
> ensures $\mathbb{P}(\mathcal{E}) \ge 1-\varepsilon$.
>
> The decision rule in Theorem1, which declares dependence if and only if $\hat{\beta} \ge \tfrac{1}{2}\log\delta$, is correct on $\mathcal{E}$ because the two competing models are separated at the population level:
> $
> \text{Null Model: } \beta = 0
> \quad\text{vs.}\quad
> \text{True Model: } \beta \ge \log\delta,
> $
> and we use the midpoint $\tfrac{1}{2}\log\delta$ as the decision threshold.
>
>
> **(e) Appendix Page 2 inequality**
>
> We have added a new lemma to justify the inequality used on page 2. The full proof is provided in **Lemma 1** in the revised appendix.
> >**Lemma (Log--Lipschitz reduction).**
> >Let $x, y > 0$ with $\min { x, y } \geq a > 0$. Then
> >$\big|\log x - \log y\big| \le \frac{1}{a}|x - y|.$
>
> **(f) Define $\alpha_{ir}$ but not use it**
>
> Good catch! In the main text, we first introduce the general SCCS likelihood with time-varying covariates $\alpha_{ir}$ for completeness (Eq.(1)), and then specialize Theorem 1 and Algorithm 2 to the case without time-varying covariates (so $r=1$ and $\alpha$ drops out). We already note right after Algorithm 2 that including $\alpha$ as a nuisance parameter would reduce efficiency and increase the required sample size. Another reason is that including it would lead to no closed-form solution for $\beta$, and a proof for that version would be highly complex. We admit this is a drawback of our results and have added more explanation and clarification after Algorithm 2 in the revised version.
>
> ### **3. PS Part**
>
> **(a) Is $\frac{\mathcal{P}}{\mathcal{P}'} = \mathcal{B}_{0.5}$?**
>
> We thank the reviewer for raising this important point. In our proof, we do not assume that $\frac{\mathcal{P}}{\mathcal{P}'} = \mathcal{B} _ {0.5}$, as such an equality would only hold in the unrealistic infinite-sample limit where $\mathcal{P}' = \mathcal{P}$. Instead, we use the estimated $\mathcal{P}'$ obtained by logistic regression; as a result we only achieve near-randomization: $\frac{\mathcal{P}}{\mathcal{P}'} \approx \mathcal{B} _ {0.5}$, with the difference controlled by
> $
> \mathbb{E} _ {X \sim \mathcal{Q}}\big|\mathcal{P}(Z=1 \mid X) - \mathcal{P}'(Z=1 \mid X)\big| \le \gamma,
> $
> where $\gamma$ decreases with sample size. Lemma 2 (Appendix B.1) further shows that the total variation distance between these distributions is $\mathcal{O}(\gamma)$. Thus, the reweighted sample behaves like randomized data up to an error term.
>
> Consequently, the proof explicitly propagates this $\gamma$ through three steps: (1) Learning $\mathcal{P}'$: Step 1 guarantees, with probability at least $1 - \epsilon/6$, that $|\mathcal{P} - \mathcal{P}'| \le \gamma$. (2) Rejection sampling: Step 2 uses Lemma 1 to show that the resulting sample is within $\mathcal{O}(\gamma)$ of the ideal randomized distribution $\mathcal{Q}\mathcal{B}_{0.5}$. (3) Final hypothesis test: Step 3 uses agnostic PAC learning results to ensure that estimating the ATE introduces at most another $\mathcal{O}(\gamma)$ error. By setting the decision threshold at $\delta/2$, we leave sufficient margin so that the total error from all steps remains below $\delta$, guaranteeing correctness with probability at least $1 - \epsilon$.

---

> > ### Author Response · Authors · 2025-10-19
> >
> > **(b) Should the sample complexity depend on $\mathcal{P}'$?**
> >
> > We thank the reviewer for this thoughtful question. The sample complexity in Theorem 2 does not directly depend on the specific fitted propensity model $\mathcal{P}'$. Instead, it depends on two key factors related to the modeling process for $\mathcal{P}'$:
> >
> > (i) the complexity of the function class used to fit $\mathcal{P}'$. The first stage of the algorithm involves learning an approximation $\mathcal{P}'$ to the true propensity function $\mathcal{P}$. By [3] and [4]  (see also the footnote on Appendix page 5), we require a sample size
> > $N_1 = \mathcal{O}\left(\frac{1}{\gamma^2}\Big(n \log \tfrac{1}{\gamma} + \log \tfrac{1}{\varepsilon}\Big)\right)$
> > where $n$ is the dimension of the covariates $X$.  This term arises from the capacity of the function class used to fit $\mathcal{P}'$, e.g., the VC dimension of a logistic regression model. If a richer model class is used (e.g., a deep neural network), the VC dimension would be larger, and thus $N_1$ would increase.
> >
> > (ii) The rejection sampling step that uses $\mathcal{P}'$ can fail if the estimated propensity scores are extremely close to $0$ or $1$. To prevent instability, we assume the standard positivity condition:
> >     $
> >     \mathcal{P}'(Z=1 \mid X) \geq \delta > 0 \text{ ,for all } X.
> >     $
> > This ensures that the sampling weights remain bounded, so that enough samples are kept after the rejection step.  A smaller $\delta$ (weaker overlap) implies that more initial samples are required before rejection in order to retain a sufficient effective sample size.
> >
> > In summary, the sample complexity depends on $\mathcal{P}'$ only indirectly, through (i) function class complexity and (ii) positivity. Once these are controlled, the subsequent proof tracks only $\gamma$ and $\delta$.
> >
> > **(c) Why is $p_{accept}$ defined as such? Where is $\mathcal{P}$? How is $p_{avg}$ formally defined? Is $p_{avg} = p_{accept}$?**
> >
> > The purpose of $p_{accept}$ is to correct for imbalance in treatment assignment by selectively accepting or rejecting samples. From standard rejection sampling theory (see section 2.3 of [5]), to convert from a source distribution $p_{source}$ to a target distribution $p_{target}$, the acceptance probability is
> > $
> > p_{\text{accept}}(X,Z) = \min\Big(c \frac{p_{target}(X,Z)}{p_{source}(X,Z)}, 1\Big),
> > $
> > so in our case
> > $
> > p_{accept}(X,Z) = \min\Big(c\frac{\mathcal{B}_{0.5}(Z \mid X)}{\mathcal{P}(Z \mid X)}, 1 \Big).
> > $
> >
> > Given $\mathcal{B} _ {0.5}(Z \mid X) = 0.5$, and since $\mathcal{P}$ is unknown, we fit $\mathcal{P}'$ to approximate it, yielding the practical rule:
> > $
> > p_{accept}(X,Z) = \min\Big(\frac{c}{\mathcal{P}'(Z \mid X)}, 1\Big),
> > $
> > where $c$ is chosen to avoid extreme weights (e.g., the median of $\mathcal{P}'(Z\mid X)$). Drawing $(X,Z)\sim \mathcal{Q}\mathcal{P}(\cdot\mid X)$, the average acceptance probability is
> >
> > $
> > p _ {avg} = \mathbb{E} _{(X,Z)\sim \mathcal{Q} \mathcal{P}}[p _ {accept}(X,Z)]
> > = \mathbb{E} _ {X \sim \mathcal{Q}} \mathbb{E} _ {Z \sim \mathcal{P}(. \mid X)} [\min\Big( \frac{c}{\mathcal{P}'(Z \mid X)},1\Big) ].
> > $
> >
> > Thus, $p_{\mathrm{avg}}$ is the expectation of $p_{\mathrm{accept}}$ under the source law $\mathcal{Q} \mathcal{P}$ and is not equal to $p_{\mathrm{accept}}$ itself. We have updated the notation and explicitly included this definition in the revised version.
> >
> > **(d) Derivation for Equation (4) in the appendix**
> >
> > Let $\mu := N_2 p_{avg}$ denote the mean number of kept samples after rejection sampling. From the appendix (Chernoff lower tail), for failure probability at most $\epsilon/3$,
> > $
> > \mathbb{P}\big(N_3 \le (1-\gamma)\mu\big)\le \exp\Big(-\frac{\gamma^2}{2}\mu\Big)
> > \quad\Rightarrow\quad
> > \gamma \ge \sqrt{\frac{2\log(3/\epsilon)}{\mu}}.
> > $
> >
> > To guarantee at least $N_3$ retained samples, we need
> > $
> > N_3 \le (1-\gamma)\mu
> > \ \Longleftrightarrow\
> > \mu - \sqrt{2\mu\log(3/\epsilon)} \ge N_3 .
> > $
> > Setting $L := \log(3/\epsilon)$ and solving the quadratic form
> > $
> > (\sqrt{\mu})^2 - \sqrt{2L\mu} - N_3 \ge 0.
> > $
> > The smallest feasible $u$ is the positive root, so
> > $
> > \sqrt\mu \ge \frac{\sqrt{2L} + \sqrt{2L + 4N_3}}{2}
> > \quad\Longrightarrow\quad
> > \mu \ge N_3 + L + \sqrt{L^2 + 2LN_3}.
> > $
> > Recalling $\mu=N_2\,p_{avg}$, we have
> > $
> > N_2
> > \ge
> > \frac{N_3 + \log(3/\epsilon) + \sqrt{(\log(3/\epsilon))^2 + 2N_3\log(3/\epsilon)}}{p_{avg}}.
> > $
> > Finally, applying the overlap lower bound $p_{avg}\ge \delta$ yields the displayed bound in the appendix. We noticed a missing constant in Equation (4) and have fixed it in the revised version.

---

> > > ### Author Response · Authors · 2025-10-19
> > >
> > > ### **4. IV Part: SEMs of the IV algorithm**
> > >
> > > For Figure 2, we follow the standard IV setting (see Figure 23.1 of [6] or section 16.1 of [7]). Below we present the just-identified, no-covariates version of IV corresponding to Algorithm 4 and Theorem 3:
> > > $$
> > > D := f_D(\varepsilon_D),\quad
> > > Z := f_Z(D, U, \varepsilon_Z),\quad
> > > Y := f_Y(Z, U, \varepsilon_Y),
> > > $$
> > > with mutually independent exogenous noises $(\varepsilon_D,\varepsilon_Z,\varepsilon_Y)$, each also independent of $U$. Assumptions:
> > >
> > > (i) Relevance: $f_Z$ depends on $D$ (i.e., $D \to Z$).
> > >
> > > (ii) Independence: $D$ is independent of the unobserved confounders $U$ (enforced by taking $D$ as a function of its own noise only).
> > >
> > > (iii) Exclusion: $f_Y$ does not take $D$ as an argument as $D$ affects $Y$ only through $Z$ (i.e., $Y(D,Z)=Y(Z)$).
> > >
> > > For the linear additive SEM commonly in 2SLS derivations [8]:
> > > $$
> > > D = \eta_0 + \varepsilon_D,\quad
> > > Z = \alpha D + \gamma U + \xi_1\ (\alpha \ne 0),\quad
> > > Y = \beta Z + \delta U + \xi_2,
> > > $$
> > > where $\varepsilon_D \perp (U,\xi_1,\xi_2)$ and all errors are mean-zero with finite variance. Note that $U$ is unobserved confounding, so it cannot be directly controlled for. Instead, it is absorbed into the error terms, which is why we distinguish $\varepsilon_Z$ from $\xi_1$ and $\varepsilon_Y$ from $\xi_2$.
> > >
> > > ### **5. General concerns: PACC and hypothesis testing**
> > >
> > > We sincerely thank the reviewer for raising this valuable point. While a single pairwise comparison in our setting may superficially resemble a hypothesis test, the PACC framework is conceptually stronger and formally more general. Below, we articulate the theoretical connections and differences between PACC Discovery and hypothesis testing, and we will add a subsection in the Related Work section for later revision to emphasize these links.
> > >
> > > **Classical hypothesis testing as a weak special case.**
> > > Classical hypothesis testing addresses a single null-alternative pair with known forms and provides Type I/II error control for that particular pair. In contrast, PACC Discovery defines a causal concept $c$ represented by a family of model pairs
> > > $
> > > \mathcal{F}_{\delta,c} =\Big( \langle \mathcal{M} _ {j1}, \mathcal{M} _ {j2} \rangle \Big) _ {j \ge 1}
> > > $
> > > where each pair differs by at least $\delta$ in causal dependence. The learner must succeed on any possible pair in this family with probability at least $1-\epsilon$, i.e.,
> > > $$
> > > \mathbb{P} _ {S \sim \mathcal{M} _ j}[\mathcal{L}(S)=\mathcal{M} _ {j,\text{true}}] \ge 1-\epsilon, \quad \forall \langle\mathcal{M} _ {j1},\mathcal{M} _ {j2}\rangle  \in \mathcal{F} _ {\delta,c}.
> > > $$
> > > This uniform quantification is strictly stronger than pointwise guarantees in classical hypothesis testing. The latter provides a one-time accept/reject rule for a fixed pair, whereas PACC Discovery studies whether a learning algorithm $\mathcal{L}$ can generalize across an entire causal family using polynomially many samples in $(1/\epsilon, 1/\delta)$. The resulting guarantees are uniform, distribution-free, and resource-aware, with properties that traditional testing typically lacks, as its guarantees depend on specific parametric forms or likelihood structures. We refer to the second point of reviewer qDwi for an extended discussion of PACC in relation to composite hypothesis testing and distribution property testing.
> > >
> > > **References**
> > >
> > > [1] H. J. Whitaker, H. Farrington, B. Spiessens, and P. Musonda. Tutorial in biostatistics: The self-controlled case series method. Statistics in Medicine, 25(10):1768--1797, 2006.
> > >
> > > [2] Lee, K. M., & Cheung, Y. B. (2024). Estimation and reduction of bias in self‐controlled case series with non‐rare event dependent outcomes and heterogeneous populations. Statistics in Medicine, 43(10), 1955-1972.
> > >
> > > [3] D. Haussler. Decision theoretic generalizations of the PAC model for neural net and other learning applications. Information and Computation, 100(1):78--150, 1992.
> > >
> > > [4] M. J. Kearns and R. E. Schapire. Efficient distribution-free learning of probabilistic concepts. Journal of Computer and System Sciences, 48(3):464--497, 1994.
> > >
> > > [5] C. P. Robert and G. Casella. Monte Carlo Statistical Methods. Springer, 2nd ed., 2004.
> > >
> > > [6] Ding, P. A First Course in Causal Inference. Chapman & Hall/CRC.
> > >
> > > [7] M. A. Hernán and J. M. Robins. Causal Inference: What If. Chapman \& Hall/CRC.
> > >
> > > [8] Grace, J. B. (2021). Instrumental variable methods in structural equation models. Methods in Ecology and Evolution, 12(7), 1148-1157.

---

### Review · Reviewer_zmtW · 2025-09-18

**Summary Of Contributions:**

In this paper authors consider the problem of causal discovery from the finite sample regime inspired by the PAC learning theory. More specifically, the authors explore the sample complexity guarantees for distinguishing which model $M_{1}$ or $M_{2}$ is the true one with probability at least $1- \epsilon$ and under assumption that the "distance" between these two models at least $\delta$. This problem was considered for three different frameworks SCCS, Propensity Score and Instrumental variable.  For these setups the "distance" between the model was defined via one of the following quantities $\delta = P(Y|Z) / P(Y|\neg Z)$ or $\delta = P(Y|Z) - P(Y|\neg Z)$.

**Additional Comments:**

N/A

**Audience:**

No

**Audience Explanation:**

My main concern for the work is about how the causal discovery problem is presented. In the classic literature, the causal discovery problem assumes that given the observational data one wants to recover the causal graph. For this problem it was shown that under the infinite data one can recover the graph up to the Markov equivalence class. And one of the challenges of causal discovery problem is to figure out the correct direction of the directed edge between two random variables.

In this work, the authors only propose a method to choose "more correct" model from the two given candidates under the assumption that one of them generated the given data. First of all, the proposed method does not construct the studied pair of the models or it does not specify how one should construct them. Further, the proposed methods only verify whether two random variables are dependent or not, but it does give us any information regarding the direction of the arrow between the random variables. Finally, the considered framework already assumes some causal structure (like instrumental variable), so it is not clear what is the actual causal discovery problem for the considered framework.

I would like to ask the authors to elaborate more on what is the exact connection of the proposed work to the classical causal discovery problem, because it seem for me to be very week and not directly related.

**Claims And Evidence:**

Yes

**Claims Explanation:**

The mathemetical proofs and statements seem to be correct and sound, however there is a chance that I can miss something.
However, I have the following concerns regarding the setup:
- One of the assumptions considered by authors for SCCS framework assumes that $Y_t$ independent of $Z_t$ given the previous historical data of assignment of treatment $Z$, covariates $X$ and outcome $Y$. I am confused, since it would be reasonably expected that the treatment $Z$ should have a direct influence on the outcome $Y$, that is they can never be independent. I would ask if the authors can specify an exact citations where such an assumption was used in the potential outcome framework or to clarify better the considered setting.
- The proposed distance function does not seem reasonable to me. For example, by the proposed distance it is already assumed that $P(Y|Z)>P(Y|\neg Z)$, and it is not clear why the inequality can not be in another direction.

**Requested Changes:**

I would ask the authors to elaborate more on the raised concerns in the comments above.

---

> ### Author Response · Authors · 2025-10-18
>
> We would like to sincerely thank the reviewer for their thoughtful and constructive feedback. We have carefully considered all comments and suggestions and have revised the manuscript accordingly. Below, we provide a detailed, point-by-point response, highlighting the changes made and clarifying our reasoning where appropriate.
>
> #### **1. Confusion about Assumption 1**
>
> The reviewer raised a concern that treatment $Z_t$ has a direct causal influence on the outcome $Y_t$, and therefore $Z_t$ and $Y_t$ can never be independent.
>
> The reviewer is correct, and we made a mistake in the subscript of $Y$: it should be $t+1$ rather than $t$. This ensures that, given full knowledge of a patient’s past history, the decision to assign treatment today is statistically independent of what the patient’s future potential outcome would be. This is consistent with standard sequential ignorability formulations in longitudinal causal inference.
>
> We refer to Section 19.4 of Hernán and Robins [1] and Assumption 29.1 in Ding [2] as exact citations where such an assumption has been used within the potential outcomes framework.
> Based on these considerations, we have revised Assumption 1 as follows:
> > **Assumption (No Unmeasured Time-Varying Confounders, NUTVC)**
> > For all time points $t$, conditional on the entire history of past exposures, outcomes, and time-varying covariates, the exposure at time $t$ is independent of future potential outcomes. Formally,
> > $$ Z _ t \perp Y _ {t+1}(\bar{z} _ t) | \bar{Z} _ {t-1} , \bar{Y} _ t, \bar{X} _ t, \quad \text{for all } t $$
> > where $\bar{z} _ t = (z _1, \ldots, z _ t)$ is a fixed realization of the exposure history, and $Y _ {t+1}(\bar{z} _ t)$ is the potential outcome at time $t+1$ under exposure history $\bar{z} _ t$.
>
>
> #### **2. Distance Function Justification**
>
> Conceptually, the distance function serves as a quantitative measure of separability between competing data-generating mechanisms. A nonzero separation (i.e., effect size) ensures that, with sufficient data, the learner can reliably favor the correct model by detecting a nontrivial gap. This parallels the logic underlying classical power analysis in clinical trials, in which required sample size depends on the hypothesized effect size and desired Type I/II error [3].
>
> Our PACC framework allows multiple types of distance so that practitioners can tailor it to the outcome type and specific task. Examples include:
>
> - Risk difference:  $\mathbb{P}(Y \mid Z)-\mathbb{P}(Y \mid \neg Z) \geq \delta$.
> - Risk (or rate) ratio margin:  $\tfrac{\mathbb{P}(Y \mid Z)}{\mathbb{P}(Y \mid \neg Z)} \geq \delta$.
> - KL divergence: $D_{\mathrm{KL}}\big(P_{Y\mid Z}\,\|\,P_{Y\mid \neg Z}\big) \geq \delta$.
>
> The reviewer mentioned the case in which we use the format $\mathbb{P}(Y \mid Z) > \mathbb{P}(Y \mid \neg Z)$ and asked why the inequality cannot be reversed. We use the current format to illustrate a common scenario in which the probability of a favorable outcome (e.g., survival) is higher when treatment is given. Conversely, if the treatment is expected to decrease the probability of an undesirable event (e.g., heart attack), the inequality can simply be reversed:
> $
> \mathbb{P}(Y \mid Z) < \mathbb{P}(Y \mid \neg Z).
> $
> Our framework fully supports both cases, and the theoretical results remain valid regardless of inequality direction.
> Moreover, for a two-tailed question that whether $\mathbb{P}(Y \mid Z)$ and $\mathbb{P}(Y \mid \neg Z)$ differ in either direction, this can be formulated in PACC by defining the competing model families as:
> $$ \mathcal{M} _ {j1} = \{ \mathbb{P}(Y \mid Z) = \mathbb{P}(Y \mid \neg Z) \},
> \mathcal{M} _ {j2} = \{ |\mathbb{P}(Y \mid Z) - \mathbb{P}(Y \mid \neg Z)| \ge \delta \}. $$
> This formulation corresponds to a two-sided alternative and is compatible with our general PACC decision rule. In short, the direction of the inequality is purely a matter of convention and does not affect the generality or correctness of our framework.

---

> > ### Author Response · Authors · 2025-10-18
> >
> > #### **3. Main Concern**
> >
> > This is an excellent point, and we appreciate the opportunity to clarify. Before addressing the reviewer’s specific concerns in detail, we first provide a high-level summary of our contribution and how it aligns with the broader goals of the modern causal inference community.
> >
> > Our intention is not to replace existing causal frameworks, but to provide a complementary finite-sample perspective. Most theoretical analyses assume infinite samples or asymptotic guarantees, which are unrealistic in practice. Our framework explicitly incorporates finite-data and resource constraints, offering principled guidance on practical questions such as: given a desired error tolerance, what sample size is required to answer a specific causal question using observational data? We further illustrate this with a new corollary **(Appendix A.2)** in the revised version showing a case where a problem is identifiable but not PACC-discoverable under finite-sample constraints.
> >
> > As the reviewer mentioned, one of the classical tasks in causal discovery is the complete recovery of the underlying causal graph. We argue that this task can be viewed as a special case of the PACC framework, obtained by pairing the correct causal graph with every possible incorrect graph. For example, the true data-generating process may be represented by a Bayesian Network, and the competing models could also be Bayesian Networks defined on the same set of variables but differing in their edge structures to account for alternative causal relationships.
> >
> > While this complete recovery problem is theoretically attractive, it is often intractable or practically infeasible. In many scientific and applied settings, the research goal is much more modest. For instance, given an unknown and potentially complex causal graph, we may only wish to answer a question such as: Does drug $Z$ influence outcome event $Y$? Even if the true causal graph contains many variables and intricate dependencies, addressing this question requires relating only the two variables of interest.
> >
> > **(a) Choosing the “more correct” model.**
> > In our framework, choosing the more correct model plays the same role as finding the target concept in PAC learning [4, 5]. The key difference is that we adapt the notion of a concept to causal settings, where it is represented by a data-generating structure (DGS) or causal model. This aligns with Pearl’s view that causal inference concerns identifying and reasoning about the underlying DGS [6]. From this viewpoint, a causal learner should aim to recover the correct DGS or, at minimum, identify the component relevant to the causal question.
> >
> > **(b) How to construct the models.**
> > When defining the PACC framework, our goal was to make it as general as possible so that it could encompass a wide range of existing causal methods. However, this generality can also make the framework appear abstract or vague. To address this concern, we illustrate the framework through three concrete examples: Self-Controlled Case Series (SCCS), Propensity Score (PS), and Instrumental Variable (IV). For each, we explicitly show how the competing models $\mathcal{M}_1$ and $\mathcal{M}_2$ are constructed.
> >
> > For example, in PS analysis, we define variables $\{X, Y, Z\}$ with joint distribution $\mathcal{D}(X, Y, Z) = \mathcal{Q}(X=x) \mathcal{P}(Z=z|X=x) \mathcal{R}(Y=y | Z=z, X=x)$. This joint distribution, together with Assumptions 6–9, explicitly characterizes the class of causal models under consideration.
> > Similar constructions apply to SCCS (based on Assumptions 1–5) and to IV analysis (based on the structural formulation in Assumption 10 and Figure 2).

---

> > > ### Author Response · Authors · 2025-10-18
> > >
> > > **(c) Only testing dependence without addressing direction.**
> > > We appreciate this insightful question. Our current paper primarily focuses on dependence questions for two reasons. First, the paper is already lengthy, and we want to keep the exposition focused.  Second, questions of the form “Does $Z$ cause $Y$?” are among the most prevalent in the causal inference literature. For example, most clinical trials ask whether a drug is effective or whether a treatment reduces the risk of an adverse outcome.
> > >
> > > However, our PACC framework is inherently flexible and can naturally be extended to address other types of causal questions. These include determining the causal direction between two variables, identifying the correct causal structure among a set of variables, and estimating causal parameters or effect sizes with precision guarantees. As the reviewer explicitly mentioned the importance of addressing causal direction, we have included a new theorem in the revised version on causal direction detection under the PACC framework in **Appendix C**.
> > >
> > > We also note that this type of question can often be addressed using the concept of Granger causality [7], which incorporates temporal information. Granger causality is based on the principle that only events occurring earlier in time can cause later events. By explicitly incorporating time into causal reasoning, it provides a natural and intuitive way to determine the direction of causal edges.
> > >
> > > **(d) Assuming some causal structure.**
> > > It is common in causal studies to incorporate prior knowledge about certain parts of the causal structure. This practice reflects the reality that, in many applied domains, researchers have access to domain expertise or well-established scientific knowledge.  The reviewer’s example of IV illustrates this point well. IV methods begin with the assumption that certain variables, or instruments, satisfy specific conditions (e.g., Assumption 10), which are standard in the literature, such as Figure 23.1 of Ding [2] or Section 16.1 of Hernán and Robins [1].
> > >
> > > Many widely used causal methods, including IV, SCCS, and PS, rely on similar types of structural pre-specifications. Within the PACC framework, these assumptions simply define the space of allowable models.  Thus, the use of pre-specified causal structures is not a limitation but rather a reflection of real-world practice.
> > >
> > > #### **References**
> > >
> > > [1] Hernán, M. A., & Robins, J. M. (2020). *Causal Inference: What If.* Chapman & Hall/CRC.
> > > [2] Ding, P. (2024). *A First Course in Causal Inference.* Chapman & Hall/CRC.
> > > [3] Chow, S.-C., & Liu, J. P. (2008). *Design and Analysis of Clinical Trials.* Wiley.
> > > [4] Kearns, M., & Vazirani, U. (1994). *An Introduction to Computational Learning Theory.* MIT Press.
> > > [5] Anthony, M., & Biggs, N. (1997). *Computational Learning Theory.* Cambridge University Press.
> > > [6] Pearl, J. (2009). *Causality: Models, Reasoning, and Inference.* Cambridge University Press.
> > > [7] Granger, C. W. J. (1969). “Investigating Causal Relations by Econometric Models and Cross-Spectral Methods.” *Econometrica,* 37(3), 424–438.

---

### Review · Reviewer_qDwi · 2025-10-06

**Summary Of Contributions:**

The paper focuses on causal inference problems in the Probably Approximately Correct (PAC) setting. The paper begins by reviewing the PAC framework and describing a general problem setting, which they call PAC Causal Discovery (or PACC Discovery), in Sections 2 and 3. In Section 4, the authors prove a PAC bound on the Self-Controlled Case Series (SCSS) method, which is an observational causal inference method for time series settings. In particular, Theorem 1 gives a sample complexity bound for differentiating whether or not the outcome event $Y$ is $\delta$-dependent on an exposure (treatment) event $Z$. In Section 5, the authors analyze two other algorithms in different settings: the propensity weighting method for causal inference in a standard potential outcomes setting (i.e., assuming ignorability, consistency, and bounded positivity), and the two-stage least squares method for instrumental variable settings.

### Strengths:
- **PAC + causal:** It is nice to see an attempt to apply the PAC framework to a range of causal problems.
- **Specific bounds:** The individual sample complexity results (Theorems 1-3) are good to have.

### Weaknesses:
- **Misuse of key terms:** Several terms that are central to the paper, such as "causal discovery" and "causal model", are used incorrectly. The broadly accepted use of the term "causal discovery" is as a synonym for "causal structure learning", which seeks to recover causal relationship from data (see [1] and [2]), but the paper focuses entirely on problems related to causal effect estimation. This misuse is carried into their definition of a causal model (Definition 2) as a probability distribution. I cannot emphasize this more strongly: **in the Pearlian framework, a causal model is not a probability distribution, it induces a probability distribution**. This distinction is core to the graphical causal framework, which this paper claims to address. See, for example, the foundational distinction between *Bayesian networks* and *causal Bayesian networks* (see [3]).
- **Inflated name for a simple problem setting:** In Definition 4 and Algorithm 1, it becomes clear that the learner is *given* the paper of models to distinguish as input to the algorithm, and the goal becomes hypothesis testing to say which model is data-generating. As the authors note, this is basically classical (two-sample) hypothesis testing, and the goal is to give a worst-case bound. This is a fairly weak (easy) problem setting, since the pair is given, as opposed to similar settings like composite hypothesis testing or distribution property testing (see e.g. [4] and [5]).
- **Misunderstanding of related work:** The authors say that [6] establishes "finite-sample guarantees for recovering the interventional Markov equivalence class", which is simply incorrect. The goal of [6] is not to recover an interventional MEC, and the paper contains no such bound. The goal of [6] is causal effect estimation: see their definition of the "PAC causal effect estimation" problem in the introduction.
- **Weak results:** The proofs of the main results (Theorems 1-3) rely on a reduction to estimation, e.g. the last steps of Algorithms 2, 3, and 4 check whether an estimated is above or below some threshold. This approach seems to ignore classical hypothesis testing results (e.g., that the likelihood ratio test is uniformly most powerful) and seems to lead to suboptimal sample complexity (e.g., in Theorem 2 when $\gamma = \frac{\delta^2}{4}$, I doubt $\frac{1}{\delta^6}$ is the correct dependence in such a simple setting as a logistic model for treatment, since Theorem 1 of [6] gives a $\frac{1}{\alpha}$ bound for an analogous positivity parameter $\alpha$).


[1] Zanga et al. (2025), *A survey on causal discovery: Theory and practice*
\
[2] Huber (2024), *An introduction to causal discovery*
\
[3] https://bayes.cs.ucla.edu/BOOK-2K/ch1-3.pdf
\
[4] https://math.arizona.edu/~jwatkins/R_composite.pdf
\
[5] Cannone (2024), *Topics and techniques in distribution testing*
\
[6] Choo et al. (2025), *Probably approximately correct high-dimensional causal effect estimation given a valid adjustment set*

**Audience:**

Yes

**Audience Explanation:**

The individual bounds should be interesting, especially if the bounds in either Theorem 1 or Theorem 3 is closer to optimal than Theorem 2.

**Broader Impact Concerns:**

No concerns

**Claims And Evidence:**

Yes

**Claims Explanation:**

The technical claims (Theorems 1-3) do appear correct, with detailed proofs, they just seem weak as noted in *Summary of Contributions*.

**Requested Changes:**

Based on the weaknesses listed in *Summary of Contributions*, I suggest the following:
- **Change title and name of framework, remove inflated claims of novelty:** The paper should use much more direct and transparent language to describe what they are doing: worst-case hypothesis testing, applied to causal hypotheses. The term "causal discovery" should be avoided.
- **Better literature review:** The authors should thoroughly familiarize themselves with distribution testing and composite hypothesis testing, and look for similar worst-case hypothesis testing results. There should be plenty of introductory examples that will give intuition (e.g., distinguishing between two coins with different bias needs fewer samples than estimating the bias).
- **Fix related work:** Thoroughly check your understanding of related work and describe it correctly.

---

> ### Author Response · Authors · 2025-10-19
>
> We sincerely thank the reviewer for the thoughtful and detailed feedback, as well as for recognizing the strengths of our submission. Below, we address each point listed under weaknesses in detail. We have also revised the paper to clarify terminology, strengthen the related work discussion, and highlight the key contributions of our work.
>
> ### **1. Misuse of Key Terms**
>
> **(a) Causal discovery vs. causal effect estimation**
>
> We thank the reviewer for raising this important point. We emphasize that we use the term causal discovery because our proposed framework is broad and encompasses both structure learning and causal effect inference. In the paper, we apply the framework to several structure-learning problems, each targeting the question of whether $Z$ causally influences $Y$ by at least a minimum detectable amount $\delta$. We clarify that $\delta$ here is not a causal effect estimate but a minimum detectable difference (or faithfulness threshold) between competing causal models. Hence, $\delta$ governs the identifiability gap between models rather than the magnitude of a causal effect.
>
> We have also added a new theorem in **Appendix C** demonstrating the utility of the PACC framework for causal direction detection, further confirming that it applies beyond effect estimation. Moreover, one of the classical goals of causal structure learning, the complete recovery of the underlying causal graph, can be expressed as a special case within our framework by pairing the true causal graph with all possible incorrect graphs. For example, if the true data-generating process is represented by a Bayesian network, the competing models may be other Bayesian networks on the same set of variables but differing in edge structures to encode alternative causal relationships. Importantly, the framework can also naturally represent causal effect estimation tasks: comparing a correct model with the true effect size against an alternative model with an incorrect effect size constitutes a special case of PACC-discoverability.
>
> In summary, the PACC framework is not limited to causal effect estimation but provides a principled and unified formulation that is applicable to a wide range of causal tasks. For this reason, we respectfully retain the term causal discovery in our title and throughout the paper.
>
> **(b) Causal model vs. probability distribution**
>
> We appreciate the reviewer’s excellent point. In earlier drafts, we defined causal models using randomized Turing machines to emphasize their role as general generative mechanisms. In later versions, following feedback from other experts, we broadened the definition to include all probability distributions, thereby capturing a wider class of causal representations beyond graphical causal models. For causal models, we consider examples such as Bayesian networks, structural equation models (SEMs), point processes, Turing machines, and other probabilistic generators. All of these can be viewed abstractly as synthetic generative mechanisms, which are formal procedures capable of generating probability distributions.
>
> However, we acknowledge that defining a causal model directly as a probability distribution introduces confusion, since it departs from the Pearlian framework as suggested. In response, we have revised Definition2 to clarify this distinction and to explicitly adopt generative semantics:
>
> > **Revised Definition 2**
> > A causal model $\mathcal{M}$ over an instance space $\mathcal{I}$ is a generative mechanism that induces a probability distribution over $\mathcal{I}$.
>
> This revision aligns our framework with Pearl’s causal semantics while retaining the theoretic generality needed for PAC-style analysis. Intuitively, as we further illustrate in Example 2 of the paper, while the probability distribution captures observable associations among variables, the causal model $\mathcal{M}$ specifies the underlying mechanisms that determine how those associations would change under interventions or counterfactual manipulations.

---

> > ### Comment · Reviewer_qDwi · 2025-10-26
> > **RE: Misuse of key terms**
> >
> > **(a) Causal discovery vs. causal effect estimation**
> >
> > Thanks for the response. I see better now the motivation for using the term "causal discovery". I'd like to comment on this point:
> >
> > > *Importantly, the framework can also naturally represent causal effect estimation tasks: comparing a correct model with the true effect size against an alternative model with an incorrect effect size constitutes a special case of PACC-discoverability.*
> >
> > My original comment that the paper "focuses entirely on problems related to causal effect estimation" is closely connected to this point. In particular, the paper considers many problems where the correct model has effect size zero; hence, by the author's own reasoning, many of the problems can be treated as causal effect estimation tasks. In other words, determining whether $Z$ causally influences $Y$ by at least $\delta$ can be treated as *either* a causal discovery task *or* as a causal effect estimation task.
> >
> > The way I see it is as follows: if we have a Venn diagram of "causal effect estimation (CEE) tasks" and "causal discovery (CD) tasks", then the original submission mostly consists of problems in the intersection. The new application to detecting directions in Appendix C would be only in the "CD tasks" circle, whereas the quoted application to checking nonzero causal effects would be only in the "CEE tasks" circle. The framework encompasses both circles and more. Hence, I still have some reservations about calling the framework "PAC Causal Discovery", since it is doesn't capture the applicability to CEE tasks; I would be much more comfortable with a name like "PAC Causal Hypothesis Testing".
> >
> > **(b) Causal model vs. probability distribution**
> > Thanks for considering this important point, and I'm much happier with the revised Definition 2, which captures the important distinction between the model $\mathcal{M}$ itself, and the probability distribution over $\mathcal{I}$ that it induces. From their response, I think the authors and I are on the same page with this distinction. However, just as a brief note for absolute clarity, I don't think it's just the case that "defining a causal model directly as a probability distribution introduces confusion": it's not just about confusion/ambiguity, but also correctness. The new definition (though somewhat informal, e.g. what is a generative mechanism) is technically correct for all the frameworks that I can think of, so I appreciate the change.

---

> > > ### Author Response · Authors · 2025-10-30
> > >
> > > **(a) Causal discovery vs. causal effect estimation.**
> > >
> > > Thanks for the insightful comments! We appreciate the common ground that we both agree that our framework encompasses both the CD and CEE circles and more, and the original submission mostly consists of problems in their intersection.
> > >
> > > However, we do not entirely agree with the statement that “the current framework fails to capture the applicability to CEE tasks.” As the reviewer explicitly noted, “determining whether $Z$ causally influences $Y$ by at least $\delta$ can be treated as either a causal discovery task or a causal effect estimation task.” So our framework does capture the applicability to CEE tasks. We admit that for continuous CEE problems, the competing causal model paradigm makes CEE tasks appear less natural within our discrete formulation. To represent them within our PAC-style framework, we can discretize the effect space into intervals (or “segments”), each corresponding to a competing causal model.
> > >
> > > As for the framework name, we appreciate the suggestion and below state the two main reasons for using the name causal discovery:
> > >
> > > (1) Hypothesis testing is only a special case of our framework. As we stated in the earlier response, standard hypothesis testing is a special case of our setting without the uniform, distribution-free, and resource-aware properties, which are the main contributions of our framework.
> > >
> > > (2) We inherit the approximate learning perspective from the standard PAC learning theory. In PAC learning, the learner aims to identify the correct target concept, and in PACC, the learner aims to “discover” the correct causal model among competing models. So we adopt a more intuitive way to describe this process. Again, this notion of discovery is different from naïve hypothesis testing, it aligns more with generalizing the PAC semantics of concept identification to the causal domain rather than reformulating the standard decision-theoretic framework of hypothesis testing.
> > >
> > > We are open to suggestions regarding the paper title or framework name. In an earlier version, we titled the paper “A Theory of the Discoverable,” inspired by Valiant’s original paper, “A Theory of the Learnable.” If the concern is primarily about the framework name, we are also open to alternatives such as Probably Approximately Correct Causal Dichotomization, Distinction, or Analysis, which may better highlight the specific way we tried to formalize our view of discovery. We also note that in earlier drafts we more strongly emphasized the connection between our framework and hypothesis testing, and we will restore and expand that discussion in the revised version.
> > >
> > > **(b) Causal model vs. probability distribution.**
> > >
> > > We appreciate you bringing this problem to our attention and are glad to know you liked the changes we made. We further revise the definition as follows to avoid the informal “generative mechanism”:
> > >
> > > >**[Revised Definition 2]**
> > > >A causal model $\mathcal{M}$ over an instance space $\mathcal{I}$ is a synthetic model that captures the underlying data-generating process and induces a probability distribution over $\mathcal{I}$.

---

> > > > ### Comment · Reviewer_qDwi · 2025-11-06
> > > >
> > > > Thanks for the response! I do think that changing the name of the framework to something like "PAC Causal Analysis" would alleviate my concerns. "Causal Discovery" is such a well-established term that I think most readers would find "PAC Causal Discovery" to be confusing, and thus have a harder time reading the paper.
> > > >
> > > > I appreciate the plan to restore and expand the discussion on the connection to hypothesis testing, as well as the change to Definition 2.

---

> > > > > ### Author Response · Authors · 2025-11-07
> > > > >
> > > > > PAC Causal Analysis it is! We will upload the revised version this weekend. Thanks again for your constructive feedback!

---

> ### Author Response · Authors · 2025-10-19
>
> ### **2. Inflated Name for a Simple Problem Setting**
>
> We sincerely thank the reviewer for raising this valuable point and for the opportunity to clarify our contribution. While a single pairwise comparison in our setting may superficially resemble a hypothesis test, the PACC framework is conceptually stronger and formally more general. Below, we articulate the theoretical connections and differences between PACC Discovery, distribution property testing, and composite hypothesis testing. We will also add a subsection in the Related Work section for later revision to emphasize these links.
>
> **Historical connection**
>
> The field of distribution property testing emerged directly from the foundational ideas of PAC learning. Leslie G. Valiant’s work on PAC learning [1] introduced the notion of learning with finite-sample, uniform guarantees. Decades later, his sons Gregory and Paul Valiant formalized distribution property testing [2] as an elegant extension of this framework, asking whether an unknown distribution possesses a certain property. Both paradigms share the same $(\epsilon,\delta)$ probabilistic guarantees and worst-case quantifiers. Our PACC framework inherits this lineage as an analogue of PAC learning in the causal domain. Like PAC learning, it requires success with high probability for any admissible data-generating process; like distribution property testing, it operates over distributions, but those distributions are induced by causal models that encode distinct structural mechanisms.
>
> **Classical hypothesis testing as a weak special case**
>
> Classical hypothesis testing addresses a single null–alternative pair with known forms and provides Type I/II error control for that particular pair. In contrast, PACC Discovery defines a causal concept $c$ represented by a family of model pairs
> $
> \mathcal{F} _ {\delta,c} = \{\langle \mathcal{M} _ {j1}, \mathcal{M} _ {j2} \rangle\} _ {j \ge 1},
> $
> where each pair differs by at least $\delta$ in causal dependence. The learner must succeed on any possible pair in this family with probability at least $1-\epsilon$, i.e.,
> $$
> \mathbb{P} _ {S\sim\mathcal{M} _ {j}}[\mathcal{L}(S)=\mathcal{M} _ {j,\text{true}}] \ge 1-\epsilon,
> \quad \forall \langle\mathcal{M} _ {j1},\mathcal{M} _ {j2}\rangle \in \mathcal{F} _ {\delta,c}.
> $$
> This uniform quantification is strictly stronger than the pointwise guarantees in classical hypothesis testing. The latter provides a one-time accept/reject rule for a fixed pair, whereas PACC Discovery studies whether a learning algorithm $\mathcal{L}$ can generalize across an entire causal family using polynomially many samples in $(1/\epsilon, 1/\delta)$. The resulting guarantees are uniform, distribution-free, and resource-aware, with properties that traditional testing lacks, as its guarantees often depend on specific parametric forms or likelihood structures.
>
> **Relation to composite hypothesis testing**
>
> Composite hypothesis testing extends single hypothesis testing to a set of distributions. While this formulation shares some resemblance to ours, PACC Discovery generalizes the principle from families of distributions to the space of causal mechanisms. Each causal model $\mathcal{M}$ represents a generative process consistent with causal assumptions such as ignorability, the no–unmeasured–time-varying–confounder (NUTVC) condition, or the exclusion restriction in instrumental variables. Conceptually, this extends testing from the probability space to the causal space. In addition, we highlight the finite-sample PAC-style guarantees in the PACC framework, which are often overlooked in testing problems. Hence, PACC Discovery does not merely restate worst-case hypothesis testing; it unifies statistical distinguishability and causal learnability within a PAC-style finite-sample theoretic framework.
>
> In summary, PACC Discovery subsumes hypothesis testing, distribution property testing, and composite testing as special cases, while extending their uniform, worst-case philosophy to causal inference.

---

> > ### Author Response · Authors · 2025-10-19
> >
> > ### **3. Misunderstanding of related work**
> >
> > We appreciate the reviewer’s careful reading. You are correct, we mistakenly mixed the intended related work with another citation. We have corrected this in the revised version as follows: [3] provides finite-sample PAC guarantees for causal effect estimation via covariate adjustment, including theoretical bounds and algorithms for identifying $\epsilon$-Markov blankets and minimal targeted adjustment sets.
> >
> > We also thoroughly reviewed all related work and corrected one citation misalignment, as well as a partially inaccurate statement about R-MAX. We thank the reviewer again for bringing these issues to our attention.
> >
> >
> > ### **4. Weak results**
> >
> > We thank the reviewer for these insightful comments. Below we clarify (a) the rationale for the different dependence on $\delta$ across the three theorems and why this difference does not imply suboptimality, and (b) the comparison to the $1/\alpha$ bound in [3].
> >
> > **(a) On the $1/\delta^6$ rate in the propensity score result.**
> >
> > We agree that the bound in Theorem 2 appears looser than those in Theorems 1 and 3. This difference arises not from analytical looseness, but from the compounded dependence introduced by the three stage structure of the propensity score proof, which jointly controls (i) model approximation, (ii) rejection sampling, and (iii) estimation.
> >
> > - Approximation of the treatment model. By [4] and Theorem 5.1 of [5], a linear probabilistic concept class requires $O(1/\delta^2)$ samples to approximate the true treatment probability $P(Z{=}1|X)$ within total variation distance $\delta$.
> >
> > - Rejection sampling. The lemma 2 in Appendix B.1 bounds the discrepancy between the ideal and approximate weighted distributions $\mathcal{Q}\mathcal{P}$ and $\mathcal{Q} \mathcal{P}'$ by $O(\epsilon/\delta)$, and the acceptance probability itself scales with the marginal positivity constant $\delta$. Ensuring a sufficient number of accepted samples thus introduces an additional $1/\delta^2$ factor.
> >
> > - ATE estimation on the adjusted sample. Applying agnostic PAC learning theory [6] to the final ATE estimation step
> > requires another $O(1/\delta^2)$ samples to achieve the desired error probability. Then combining all three stages yields an overall complexity of $O(1/\delta^6)$.
> >
> > In contrast, the SCCS and IV analyses each involve direct estimation of a single contrast parameter (e.g., the log ratio or a regression slope) and therefore require only one concentration bound, yielding the optimal $O(1/\delta^2)$ rate.
> > We emphasize that we do not ignore classical hypothesis testing results: Theorems 1 and 3 reflect the standard $O(1/\delta^2)$ scaling that corresponds to the optimal likelihood ratio rate. The higher order dependence in Theorem 2 arises only because the propensity score procedure must propagate approximation error across multiple stochastic mappings.
> >
> > We acknowledge that tighter constants for propensity score methods may be achievable through refined concentration or by replacing rejection sampling with inverse propensity weighting, which we plan to explore in future work.
> >
> > **(b) Comparison to the $1/\alpha$ rate in [3].**
> >
> > We appreciate the comparison with [3]. Their $1/\alpha$ dependence arises in a different setting: finite sample causal effect estimation under a valid adjustment set, where $\alpha$ controls covariate overlap and sparsity. Our $\delta$ parameter instead quantifies the minimal detectable causal contrast (a faithfulness margin) between competing models. In short, $\alpha$ represents identifiability through sufficient treatment overlap, whereas $\delta$ captures detectability through minimal separation between causal models. While both analyses adopt PAC-style analyses, [3] address estimation accuracy of a fixed parameter, whereas our framework provides distribution free, family-wise uniform guarantees
> > over a class of causal models. Consequently, $\alpha$ and $\delta$ govern distinct sources of difficulty, and the corresponding rates are not directly comparable.
> >
> > **References**
> >
> > [1] L. G. Valiant, “A Theory of the Learnable,” Communications of the ACM, 1984.
> >
> > [2] G. Valiant and P. Valiant, “Automatic Inequalities for Probability Distributions,” Foundations of Computer Science, 2017.
> >
> > [3] Choo, D., Squires, C., Bhattacharyya, A., & Sontag, D. (2024). Probably approximately correct high-dimensional causal effect estimation given a valid adjustment set. arXiv preprint
> >
> > [4] D. Haussler. Decision theoretic generalizations of the PAC model for neural net and other learning applications. Information and Computation, 100(1):78--150, 1992.
> >
> > [5] M. J. Kearns and R. E. Schapire. Efficient distribution-free learning of probabilistic concepts. Journal of Computer and System Sciences, 48(3):464--497, 1994.
> >
> > [6] Kearns, M. J., Schapire, R. E., & Sellie, L. M. (1992, July). Toward efficient agnostic learning. In Proceedings of the fifth annual workshop on Computational learning theory (pp. 341-352).

---

> > > ### Comment · Reviewer_qDwi · 2025-10-26
> > > **RE: Misunderstanding of related work and Weak Results**
> > >
> > > ## Misunderstanding of related work
> > >
> > > Thank you for making these corrections and thoroughly reviewing the related work.
> > >
> > > ## Weak Results
> > >
> > > **(a) On the $1/\delta^6$ rate in the propensity score result**
> > > I agree that the difference may not arise from analytical looseness (though, without closely revisiting the paper, I'm curious as to why the bounds are multiplied rather than added), my main point was that this algorithm may be highly suboptimal for the task - I'm reminded of the Vapnik quote:
> > > > “When solving a problem of interest, do not solve a more general problem as an intermediate step”.
> > > I understand that the result is just one of several in the paper, so it's not reasonable to expect the absolute best bound, but I'd appreciate if the paper emphasized the potential suboptimality of this bound and perhaps even gave a conjecture as to the optimal bound based on the likelihood ratio.
> > >
> > > **(b) Comparison to the $1/\alpha$ rate in [3]**
> > > I think there is some confusion about $\delta$ and $\alpha$. Prior to Theorem 2, the authors state that $\delta = \min \{ \delta_1, \delta_2 \}$, where $\delta_1$ in Assumption 7 is a positivity parameter. Similarly, $\alpha$ in the Choo et al. paper is a positivity parameter (Equation (2) of that paper). So, when $\delta_1 < \delta_2$, these parameters play the same (or at least, very similar) roles. In Choo et al., the goal is to estimate the ATE up to some precision $\varepsilon$. In this paper, the strategy for hypothesis testing is to estimate the ATE up to some precision that depends on $\delta_2$ and $\epsilon$, then check against the threshold $\delta_2/2$. So, while the rates are not directly comparable, there should be a way to straightforwardly relate them.

---

> > > > ### Author Response · Authors · 2025-10-30
> > > >
> > > > **Inflated Name for a Simple Problem Setting**
> > > >
> > > > Thanks for the great suggestions! We are working on a more comprehensive and detailed related work section to incorporate other frameworks and more recent works as suggested.
> > > >
> > > > **Weak results**
> > > >
> > > > (a) On the $1/\delta^6$ rate in the propensity score result
> > > >
> > > > You are right that while our framework aims to solve a more general problem, the particular PS results do show suboptimal rates compared with our other results. We will emphasize this as the potential suboptimality of the sample bound and state the analytic reason behind it. As for the multiplicative bound, note that the error or uncertainty in the three stages is not independent. Each stage introduces a scaling factor that compounds the uncertainty from the previous one, and thus the multiplicative dependence ultimately gives rise to the $\mathcal{O}(1/\delta^6)$ sample complexity.
> > > >
> > > > (b) On the relation between $\alpha$ and $\delta$.
> > > >
> > > > Good instinct! In a loose comparison, $\alpha \approx \delta_1$, the positivity part of our $\delta$. However, our $\delta$ additionally includes the effect-size threshold $\delta_2$ to ensure that the competing causal models are separated not only by overlap but also by a minimal causal difference.
> > > > Therefore, $\alpha$ and $\delta$ play analogous roles in controlling identifiability, with $\alpha$ governing overlap and $\delta$ extending this notion to include causal effect distinguishability.
> > > > When $\delta_1 < \delta_2$, the bound is dominated by the overlap condition and coincides with the role of $\alpha$; when $\delta_2 < \delta_1$, the minimal effect threshold determines the distinguishability of the causal models within the PACC framework.

---

> > ### Comment · Reviewer_qDwi · 2025-10-26
> > **RE: Inflated Name for a Simple Problem Setting**
> >
> > I appreciate the connection to these other areas and the distinction between a family of distributions (in composite hypothesis testing) and a family of causal models (in the proposed framework).
> >
> > I think the work would be significantly strengthen by discussing these other frameworks in more detail, including equations for the types of guarantees given by each framework, and a formal discussion of how the proposed framework subsumes other frameworks as special cases.
> >
> > I would also strongly encourage the authors to delve into more recent work in distribution property testing: while the original Gregory/Paul Valiant definition may consider only a single distribution, more recent work on areas like distribution closeness testing considers multiple distributions, so these areas are more similar to the proposed framework.

---

### Decision · Action_Editor_Yoow · 2025-12-18

**Recommendation:** Reject

**Additional Comments:**

Some comments from the final discussion:
> In the standard PAC framework, we draw i.i.d. samples from an underlying distribution and aim to output, with high probability, a "good" hypothesis from a concept class. Importantly, there are typically multiple hypotheses to choose from. The framework also allows for improper learning, where the true data-generating distribution is not part of the concept class, and the goal is to find a hypothesis that approximates it closely. In contrast, this paper considers a much narrower setting: only two possible hypotheses, with the assumption that the data-generating function corresponds exactly to one of them. In PAC terminology, this can be viewed as a special case of proper learning with a concept class of size two. However, this framing feels somewhat artificial. It might be clearer and more appropriate to present the work as addressing a hypothesis testing problem rather than positioning it as a contribution to PAC learning per se.

> in the assumption that the models are different, on top of the assumption on the difference of causal graphs there is also an assumption on the difference of causal mechanisms that generate the distribution over the given graph, which is a very strong assumption

**Audience:**

Yes

**Audience Explanation:**

This is certainly of interest to some individuals in the TMLR audience.

**Claims And Evidence:**

No

**Claims Explanation:**

Reviewers remain concerned regarding the positioning of the paper and some of the terminology choices. While the authors made significant strides in addressing these issues during the revision and discussion period, during the final discussion there was a consensus amongst reviewers that not all concerns had been addressed and another round of review was needed. Among the central concerns was the framing of the results as "causal discovery" and/or "PAC learning"; this particular choice of framing seemed to raise doubts and confusion during review.

**Resubmission Of Major Revision:**

The authors may consider submitting a major revision at a later time.

---

> ### Author Response · Authors · 2026-01-02
>
> We appreciate all the time, attention, and expertise of the action editor and all three reviewers.  We were surprised by the decision — we felt that two of the three reviewers started out positive, and that the third reviewer also came to agree with our changes and replies over the course of the discussion.
>
> We realize a new paradigm is hard to publish and deserves pushback in review; the pushback here is about the binary decision in the paradigm.  Many broad problem types are best captured formally by binary paradigms.  Optimization problems are cast as binary decision problems in NP-completeness theory.  The most effective way to prove many concept classes are PAC-learnable is to show they are EQ-learnable, where an equivalence query (EQ) oracle gives a binary answer of "yes" or a counterexample.  Such binary oracles are not common in nature, but they yield useful PAC algorithms, and PAC captures learning in nature.  In the same way, we argue that an effective way to build algorithms that can find data generators (causal models) in nature is to build an algorithm that correctly makes binary distinctions between data generators, provided it works for all possible binary distinctions of interest.  This use of binary distinction is a formal tool analogous to binary decision problems and EQ oracles.